# Nutraceutical Role of Polyphenols and Triterpenes Present in the Extracts of Fruits and Leaves of *Olea europaea* as Antioxidants, Anti-Infectives and Anticancer Agents on Healthy Growth

**DOI:** 10.3390/molecules27072341

**Published:** 2022-04-05

**Authors:** Eva E. Rufino-Palomares, Amalia Pérez-Jiménez, Leticia García-Salguero, Khalida Mokhtari, Fernando J. Reyes-Zurita, Juan Peragón-Sánchez, José A. Lupiáñez

**Affiliations:** 1Department of Biochemistry and Molecular Biology I, Faculty of Sciences, University of Granada, Avenida Fuentenueva, 18071 Granada, Spain; evaevae@ugr.es (E.E.R.-P.); elgarcia@ugr.es (L.G.-S.); ferjes@ugr.es (F.J.R.-Z.); 2Department of Zoology, Faculty of Sciences, University of Granada, Avenida Fuentenueva, 18071 Granada, Spain; calaya@ugr.es; 3Department of Biology, Faculty of Sciences, Mohammed I University, Oujda BP 717 60000, Morocco; khalidafadoua1@gmail.com; 4Department of Experimental Biology, Biochemistry and Molecular Biology Section, Faculty of Experimental Biology, University of Jaén, 23071 Jaén, Spain; jperagon@ujaen.es

**Keywords:** animal nutrition, diet, feeding, hydroxytyrosol, maslinic acid, nutraceutical, olive tree, phytochemicals, polyphenols, triterpenes

## Abstract

There is currently a worldwide consensus and recognition of the undoubted health benefits of the so-called Mediterranean diet, with its intake being associated with a lower risk of mortality. The most important characteristics of this type of diet are based on the consumption of significant amounts of fruit, vegetables, legumes, and nuts, which provide, in addition to some active ingredients, fiber and a proportion of vegetable protein, together with extra virgin olive oil (EVOO) as the main sources of vegetable fat. Fish and meat from poultry and other small farm animals are the main sources of protein. One of the main components, as already mentioned, is EVOO, which is rich in monounsaturated fatty acids and to a lesser extent in polyunsaturated fatty acids. The intake of this type of nutrient also provides an important set of phytochemicals whose health potential is widely spread and agreed upon. These phytochemicals include significant amounts of anthocyanins, stilbenes, flavonoids, phenolic acids, and terpenes of varying complexities. Therefore, the inclusion in the diet of this type of molecules, with a proven healthy effect, provides an unquestionable preventive and/or curative activity on an important group of pathologies related to cardiovascular, infectious, and cancerous diseases, as well as those related to the metabolic syndrome. The aim of this review is therefore to shed light on the nutraceutical role of two of the main phytochemicals present in *Olea europaea* fruit and leaf extracts, polyphenols, and triterpenes, on healthy animal growth. Their immunomodulatory, anti-infective, antioxidant, anti-aging, and anti-carcinogenic capabilities show them to be potential nutraceuticals, providing healthy growth.

## 1. Introduction

The quality of life of any living being is based on two key elements: on the one hand, survival, which is qualitative, and on the other hand, longevity, which is quantitative. Both properties are intimately linked to the quality of food, which is capable of providing living beings with a series of bioactive effects related to those organic capacities that favor both survival and longevity, and which are capable of affecting capacities such as those related to adequate growth, immunological, antiviral and antibacterial, antioxidant, as well as all those that are capable of preventing and/or eliminating a series of diseases such as cardiovascular, cancer and those related to the metabolic syndrome.

From a physical-biological point of view, living beings are open systems whose functioning depends on the fulfillment of all the physicochemical, biological, genetic, and evolutionary laws that sustain them. However, the main condition that ensures vital functioning is the capacity they must have to reach and maintain, in each situation, a steady state of dynamic equilibrium (SSDE) between the environment and the organism itself. This SSDE is based on two key elements, one exogenous, the food that comes from outside, and the other endogenous, the metabolic machinery that processes it and allows the elimination of waste in an appropriate way. Any external or internal cause that leads to a partial or total disruption of this dynamic stationary flow means a significant reduction in the survival of the living organism. Therefore, a “good” diet will be a key point to ensuring both a quality life and greater survival. The inclusion in the diet of all those bioactive elements that favor the maintenance of the SSDE will also favor adequate and healthy growth together with the prevention of all those diseases that afflict all living beings, such as cancer, diabetes, infectious diseases, and those related to the cardiac and vascular systems.

Proper nutrition, as the cornerstone of preventive health, should be based on a balanced composition in which the primary elements, carbohydrates, lipids, and proteins, as energy and plastic components, as well as secondary components such as vitamins, amino acids, and essential lipids, together with a series of phytochemicals such as flavonoids, polyphenols, and terpenoids of high biological activity, all capable of providing those bioactivities necessary to achieve greater survival and longevity, are present in the right quantities. This special report aims to provide new insights into nutritional assessment and possible modifications of nutritional behaviors and supplementation to prevent time-associated disorders in order to increase life expectancy in different populations of humans and animals [1,2,3,4,5].

In this regard, the polyphenolic constituents of the olive tree, represented mainly by hydroxytyrosol, luteolin, oleuropein, verbascoside, the acids coumaric, gallic, caffeic, vanillic, and the aldehyde oleocanthal, as well as the triterpenoid components, represented mainly by maslinic, oleanolic, ursolic, and betulinic acids and by triterpene dialcohols, erythrodiol, uvaol, etc., most of which are present in both the fruits and leaves of the olive tree (*Olea europaea*), are two of the main components of the Mediterranean diet, capable of providing all the bioactivities that will be mentioned below, and whose purpose is to increase the quality of life of the organisms that consume them [1,2].

For centuries traditional medicines, in most civilizations, have used extracts of terrestrial plant plants and aquatic organisms as a source of specific chemical compounds that were capable, in many cases, of reducing and, in others, of eliminating the effects of the diseases that were being fought. Natural products play an important role in the discovery of a multitude of potentially bioactive drugs since there are a large number of studies that demonstrate their analgesic, hepatoprotective, anti-inflammatory, antimicrobial, antioxidant, and antiallodynic effects. This is why natural compounds, in one case, such as polyphenols and, in another, such as triterpenoids, have been used from traditional medicine to the present due to their bioactive potential [1,2,3,4,5].

The use of plant extracts for nutritional and therapeutic purposes is widespread. The plant origin of these products is very much in line with people’s desire to be able to cure themselves with natural medicines; this aspect, together with the efficacy and regulation possibilities, is the basis for the widespread modern use of medicinal plants.

Traditional uses as well as its novel biological effects provide us with an extraordinarily large number of different compounds which can constitute a formidable therapeutic potential. However, there are still stumbling blocks with the procurement of plant extracts with non-standardized extraction methods and a lack of definition and extraction methods, and finally with an undefined and non-repeatable qualitative and quantitative composition.

However, and in the face of these difficulties, at present, part of these difficulties can be avoided and the main phytochemicals mentioned above can be safely provided, in the appropriate amounts, in order to be able to supplement them to the diet of different types of pets with the objective of increasing their survival and longevity and with it the quality of their lives. Next, the main bioactivities of these phytochemical components of the olive tree will be analyzed, accompanied by the bibliographic citations that support them.

Although the phenolic and terpenoid composition of the olive tree varies widely, from a quantitative point of view, there are two predominant ones; so in this analysis, we are going to focus mainly on these two phytochemicals, one corresponding to the polyphenol group, hydroxytyrosol, and the other corresponding to the pentacyclic triterpene group, maslinic acid. However, most of the effects discussed for these compounds can also be extended to the other examples of each group. Figure 1 shows a list of the main bioactivities of these phytochemicals present in olive tree extracts (fruits and leaves).

## 2. Biological Activities of the Polyphenolic Components of the Olive Tree

The main polyphenolic components present in the fruit and in the wax that covers the olive leaves are represented by hydroxytyrosol, luteolin, oleuropein, verbascoside, gallic acid, vanillic acid, caffeic acid, and the aldehyde derivative oleocanthal (Figure 2).

Among all of them, the main and most studied representative is hydroxytyrosol (HT), (2-(3,4-dihydroxyphenyl) ethanol, 3,4-dihydroxyphenethyl alcohol, DOPET, C_8_H_10_O_3_, and presents a molecular mass of 154.16. It is a natural compound, whose structure corresponds to a type of polyphenol, widely distributed throughout the plant kingdom, being especially abundant in the Oleaceae family. It can be found in high concentrations in extra olive oil, fruits and leaves of *Olea europaea*, and in other products such as fruits, vegetables, and tea. HT in oil is found in free form, as an acetate, or as part of more complex compounds such as oleuropein (a secoiridoide glycoside esterified with a phenylpropanoid alcohol, the major phenolic component of green olive pulp), the flavonoid luteolin (a 5,7-dihidroxi-4-cromenona with hydroxytyrosol), and verbascoside (a phenylethanoid with hydroxytyrosol and a phenylpropanoid sugar ester with caffeic acid).

Oleuropein and oleocanthal are responsible for the bitter taste of olives and decrease as the fruit ripens to a non-glycosylated form, oleuropein glycone by enzymatic hydrolysis, and finally converted to hydroxytyrosol. HT is a white powder with a melting point of about 55 °C and is fairly soluble in water and polar organic solvents such as low molecular weight alcohols. Like other polyphenols, it is easily oxidized in aqueous solutions unless proper precautions are taken, acidic pH and the absence of oxidants [2]. As a consequence of its polar character, HT is found in large quantities in the residues of oil processing, such as olive pomace oil, olive mill wastewater, or rinsing water. For this reason, by-products of olive oil production are an important source of hydroxytyrosol [3].

HT can be isolated from olive solid wastes during olive oil production in food industries, ameliorating the potential environmental pollution problem of the by-products. The bioconversion of these wastes into useful products is receiving increasing attention and the production of HT in this process is among the most economically relevant components of the phenolic fraction of olive solid wastes, being used as a promising natural compound for its nutritional and pharmacological importance.

This small molecule is a potent inhibitor of copper sulphate-induced LDL oxidation and platelet aggregation (potentially reducing the risk of coronary heart disease and atherosclerosis), acts as an antioxidant and metal chelator in animal models, removes superoxide anion (O_2_^•−^) generated in cells or the xanthine/xanthine oxidase system, inhibits proliferation and induces apoptosis in cancer cells, prevents DNA damage and peroxynitrite-induced tyrosine nitration, protects against hydrogen peroxide (H_2_O_2_)-induced damage, inhibits oxidative stress induced by passive smoking, exhibits anti-inflammatory activity, exerts antimicrobial and antimycoplasma activity, among other properties [2,4,5,6].

The main molecular pathways involved in the anticancer and immunomodulatory activities of hydroxytyrosol are shown in Figure 3. In addition, the main cell lines and living organisms used in this study, together with the target cells and molecular markers for the bioactivity of the polyphenols present in olive extracts, are shown in Table 1.

### 2.1. Pharmacokinetics and Toxicity

In relation to pharmacokinetics, bioavailability studies have shown that HT is absorbed in animals and humans in a dose-dependent manner but following non-lineal kinetics, after ingestion of olive oil, exerts its biological effects and is excreted in the urine mainly as glucuronide conjugates [7,8]. Absorption of HT takes place in the small intestine and colon [9]. It has been suggested that transport across the intestinal epithelium may occur by passive bidirectional diffusion [10]. Absorption of this compound is rapid, reaching peak plasma concentrations 5–10 min after ingestion, followed by a rapid decline [11]. This absorption differs according to the vehicle in which it is transported, with several authors demonstrating that rats absorbed 75% of HT when administered in an aqueous solution and 90% when an oily vehicle was used [12].

Different authors [13] have studied the tissue distribution of intravenously administered ^14^C-labeled HT in rats. At 5 min after injection, less than 8% was still present in the blood, approximately 6% in the plasma, and 1.9% in the cellular fraction of the blood; therefore, it was estimated that the half-life of the compound in blood was very small, below 2 min. Similar levels of ^14^C were found in the cells of skeletal muscle, liver, lungs, and heart, while the kidney cells accumulated 10 times more than the other organs. ^14^C was also detected in the brain, indicating that HT was able to cross the blood–brain barrier, although it has been described that HT can also be generated endogenously in the brain from dopamine [14] and from dihydroxyphenylacetic acid (DHPA) via dihydroxyphenylacetic reductase present in the brain [15].

HT metabolites were found in blood 5 min after intravenous injection of ^14^C-HT, indicating that the compound was rapidly metabolized in cells, especially in the liver and enterocytes, proposing three metabolic pathways for HT: oxidation, through the enzymes alcohol-dehydrogenase and aldehyde-dehydrogenase giving rise to dihydroxyphenylacetic acid; methylation, through the enzyme catechol-*O*-methyltransferase (COMT) giving rise to homovanillic alcohol (HVOL); and methylation plus oxidation, to form homovanillic acid (HVA) [13].

Regarding excretion, it was found that 90% of the radioactivity was detected in the urine 5 h after the intravenous injection of ^14^C-HT and a small proportion was excreted in the feces [13]. These results coincide with the results obtained in humans where most of the HT and tyrosol were found in the urine collected during the first 4 h after administration. ingestion of 50 mL of virgin olive oil [16].

Serra et al. [17], using Wistar rats as an in vivo model, showed that just one hour after ingestion of a phenolic extract from olive pomace, these phenolic compounds and a large part of their metabolic derivatives were absorbed, metabolized, and distributed throughout the body. Most tissues such as the liver, heart, spleen, thymus, testicles, and even the brain after crossing the blood–brain barrier. In addition, they revealed a clear renal route of detoxification [17].

The safety profile of HT appears to be excellent; no adverse effects have been demonstrated even at very high doses [13,18,19]. To study acute toxicity, a single dose of 2 g/kg body weight was administered to rats and no toxic effects or macroscopic alterations were found in organs; only the appearance of piloerection was reported 2 h after administration, and this disappeared in less than 48 h [13]. In addition to this phenomenon, several toxicity studies have been conducted using aqueous extracts of olive pulp in which the HT content ranged from 50–70% of the total amount of phenols [19].

In another study, oral administration to Sprague-Dawley rats of a single gavage dose of solid olive pulp extract at levels between 0 and 2 g/kg caused no adverse effects, except in soft or liquid feces [18]. At a dose of 2 g/kg/day of this extract, no acute toxicity was found, with no teratogenic or mutagenic effects. As part of a micronucleus assay, Sprague-Dawley rats were given a single dose of 5 g olive pulp extract/kg by gavage and after 6 days were given the same daily dose for the next 29 days with no mortality or clinical signs of toxicity. This study showed that the LD_50_ of the solid olive pulp extract was greater than 5 g/kg (equivalent to 3 g/kg HT), suggesting that the extract is practically nontoxic [18]. Given its excellent safety profile, future availability as a health food supplement could be envisaged.

### 2.2. Antioxidant and Anti-Ageing Effects

Defense against reactive oxygen species (ROS) is essential to protect cellular molecules such as lipids, proteins, or DNA and to prevent the development of damage that can lead to major diseases. When defensive mechanisms are overcome by the action of free radicals, subsequent cellular damage can lead to diseases such as atherosclerosis, cardiovascular, skin, and neurodegenerative diseases, diabetes mellitus, metabolic syndrome, and cancer, among others. Finally, physiological processes such as aging have been associated with an imbalance between the action of ROS and antioxidants. Some antioxidant agents can be found in various types of food.

Hydroxytyrosol (HT), compared to other olive oil phenolic compounds, shows much more effective antioxidant characteristics, such as scavenging free radicals, breaking peroxidative chain reactions, preventing lipid peroxidation, inhibiting hypochlorous acid-derived radicals, etc. [5]. In addition, a decrease in ROS production, derived from iron- or copper-induced oxidation of low-density lipoprotein (LDL), has been reported after treatment with HT in an in vitro model, suggesting a chelating action on these metals [20]. The ability to scavenge or reduce ROS generation was further confirmed in both 12-myristate 13-phorbol acetate (PMA)-treated leukocytes and in the hypoxanthine/xanthine oxidase cell-free system by chemiluminescence methods [21,22].

In a recent study, Hybertson et al. [23] demonstrated the antioxidant effects of a mixture of phenols formed carnosol, withaferin A, and luteolin via the nuclear factor erythroid 2 (Nrf2)-mediated pathway. Separately, each of these polyphenols exhibited significant antioxidant activity based on the ability to activate Nrf2 by binding to different antioxidant response elements (ARE), facilitating the regulation of the expression of a wide variety of cytoprotective genes [23].

In addition, this pathway is synergistically activated by the combined presence of the three polyphenols [22]. Their activity on canonical genes, non-canonical genes, as well as genes that appear to be regulated by mechanisms other than Nrf2 in cultured HepG2 cells, was able to protect these cells from oxidative stress challenge caused by the presence of different oxygen-free radicals such as peroxides [23].

### 2.3. Anticancer and Antiproliferaty Effects

Nowadays, cancer is among the deadliest diseases. Research has shown that the incidence of several cancers is much lower in the Mediterranean area compared to other countries. Thus, the consumption of extra virgin olive oil may be associated with a reduced risk of certain cancers, a fact that has been supported by several epidemiological studies in humans [24]. These health benefits may be attributed more to the phenolic compounds than to their fatty acid profile, as several studies have suggested that polyphenols may have a protective effect against cancer [25,26].

The HT-rich diet has recently received special attention due to its antioxidant, anti-proliferative, pro-apoptotic, and anti-inflammatory activities, capable of specifically counteracting all the hallmarks of cancer. Thus, mounting experimental evidence has suggested that, in addition to antioxidant and anti-inflammatory capacities, HT exerts anticancer effects through the activation of molecular signaling pathways, leading to cell apoptosis and growth arrest in several tumor cell lines [4,27,28,29].

It is well known that H_2_O_2_ plays a role in cell signaling, and this fact seems to be related to proapoptotic effects [30]. Furthermore, proapoptotic effects were observed in HL60 cells incubated with HT even under conditions that do not support H_2_O_2_ accumulation (between 23.8 and 38.0% depending on the medium) suggesting that mechanisms other than H_2_O_2_-releasing activity could be involved in proapoptotic activity, probably due a mitochondrial apoptotic way [31]. Polyphenols such as HT could therefore be beneficial for their contribution to redox homeostasis. On the other hand, several studies have shown that HT is able to arrest the cell cycle, reduce growth and proliferation and induce apoptosis in HL60 and HT29 cells [32].

Other authors also studied the behavior of HT at concentrations of 50–100 μM on the cell cycle in the HL60 cell line, demonstrating inhibition of cell proliferation, arresting cells in the G_0_/G_1_ phase with a concomitant decrease in the S and G_2_/M phases [33]. Subsequently, the same authors showed that HT (100 μM) causes an increase in the expression of p21^WAF/Cip1^ (cyclin-dependent kinase inhibitor 1A) and p27^Kip1^ (cyclin-dependent kinase inhibitor 1B) and inhibits cyclin-dependent kinase-6 in the same cell type, arresting the G_2_/M and G_0_/G_1_ phases of the cell cycle; they also found that HT promotes apoptosis in cells that are in S phase [34].

At the same time, different authors showed in MCF-7 breast cancer cells that HT and olive leaf extract at different doses exhibited a blockade of the G_1_ to S phase transition, manifested by an increase in the number of cells in the G_0_/G_1_ phase [28,33]. Similarly, both HT and olive leaf extract rich in HT and oleuropein decreased the number of cells in the G_2_/M phase of the cell cycle [28,35].

Recently, the molecules of luteolin have been described as a potential agent in triple-negative breast cancer (MDA-MB-231 cell line) treatments by suppressing the protein expressions of nuclear erythroid factor 2 (Nrf2), heme oxygenase 1 (HO-1), and Crypto-1, which contribute significantly and critically to the functioning of cancer stem cells [36]. Luteolin is capable of inhibiting the expressions of stem-related transcription factors, the ATP-binding cassette transporter G2 (ABCG2), CD44, the activity of aldehyde dehydrogenase 1 (ALDH1), as well as the spherical-forming properties of breast cancer stem cells [36].

In this regard, other authors studied the anticancer and antiangiogenic activity of another of the polyphenols present in olive extracts, oleocanthal, in hepatocellular carcinoma, demonstrating not only inhibition of proliferation but also inhibition of cell cycle progression and cell death, cell cycle progression, and induction of apoptosis by affecting the cell cycle protein cyclin D1 and the anti-apoptotic proteins Bcl-2 and survivin, but was also able to inhibit epithelial–mesenchymal transition (EMT) through down-regulation of Twist, which is a direct target of STAT3 (signal transducer and activator of transcription 3) together with down-regulation of MMP2 [37].

Overall, the data provided in these studies are positive in terms of the anti-tumor and anti-angiogenic capacity of the different olive polyphenols in different cancer cells, supported by studies of the inhibition of cell proliferation, anti-metastatic potential, and decreased expression of certain angiogenic factors. Although not conclusive, they indicate that polyphenols have properties and potential therapeutic effects that could be taken into account for the description of the possible beneficial effects of olive oil and the development of supplements based on polyphenolic extracts and nutraceuticals.

### 2.4. Immunomodulatory and Anti-Inflammatory Effects

HT has been described as the most powerful anti-inflammatory compound among the polyphenols present in olive oil, showing an effective inhibition of NO and prostaglandin E2 (PGE2) production, decreasing the secretion of cytokines (IL-1α, IL-1β, IL-6, IL-12, TNF-α) and chemokines (CXCL10/IP-10, CCL2/MPC-1), and reducing gene expression of iNOS, IL-1α, CXCL10/IP-10, MIP-1β, matrix metalloproteinase-9 (MMP-9), and prostaglandin E2 synthase (PGES) [38,39,40,41].

In vitro studies in human monocytic THP-1 cells treated with LPS to induce inflammatory response have shown that HT showed inhibition of proinflammatory cytokines (TNF-α) and reduced cyclooxygenase-2 (COX-2) and inducible nitric oxide synthase (iNOS) expression by more than 60% [38]. Other studies showed that HT decreased thromboxane B2 (TXB2) synthesis in an in vitro platelet-rich plasma model, probably as a result of reduced production of arachidonic acid-derived eicosanoids [42]. In this regard, these authors have found a reduction in thromboxane A2 (TXA2) synthesis as measured by the reduction of its metabolite, thromboxane B2. This decrease was mainly due to the inhibition of the activity of the enzyme cyclooxygenase [43,44].

In addition, HT prevents prostaglandin E2 synthesis by indirectly blocking iNOS and COX-2 enzymes. This effect arises from the prevention of transcriptional activation of NF-κB, interferon regulatory factor-1, and transducer and activator of transcription 1a, which prevent activation of J774 mouse macrophages [45].

HT is known to be capable of producing arylating/alkylating adducts on NF-κB cysteine residues. The action of HT on this factor blocks the transcription of COX-2 and 5-lipoxygenase enzymes, reducing prostaglandin E2 synthesis and thus the chronic influence associated with diseases such as cancer [46].

In vivo studies in rats with acute inflammation induced by the treatment through interplantar injection of different doses of carrageenan showed that rodents that received, by gavage feeding, a preparation in which the main ingredients included HT (22%), showed significant inhibition of both acute inflammation and pain associated with carrageenan administration [47]. They also found beneficial effects in patients with stabilized coronary heart disease [48].

Intake of virgin olive oil, rich in HT, was shown to be more effective than refined olive oil in reducing levels of interleukin-6 (IL-6) and C-reactive protein (CRP), recognized inflammatory markers in cardiovascular disease [49].

### 2.5. Antiviral and Antimicrobial Effects

Different studies have shown that olive oil and olive leaf extracts act as antimicrobial agents with activity against *Escherichia coli*, *Candida albicans*, *Kluyveromyces marxianus*, *Clostridium perfringens*, *Streptococcus mutans*, *Shigella sonnei*, *Salmonella enterica*, and others. It appears that the main components of olives and olive leaves responsible for the antimicrobial effect are HT and the dialdehyde and decarboxymethyl forms of elenolic acid [50,51,52].

In addition, it has been shown in vitro that HT at low concentrations also possesses antimicrobial properties against infectious agents of the respiratory and gastrointestinal tract such as *Vibrio parahaemolyticus*, *Vibrio cholerae*, *Salmonella typhi*, *Haemophilus influenzae*, *Staphylococcus aureus*, *Moraxella catarrhalis*, and are also effective against mycoplasmas such as *Mycoplasma pneumoniae* [52,53,54]. These concentrations are even lower than those used with certain antibiotics, such as ampicillin.

Yamada et al. demonstrated that hydroxytyrosol (HT) was able to inactivate different influenza A viruses, including H1N1, H3N2, H5N1, and H9N2, and also the Newcastle disease virus, suggesting that the mechanism of the antiviral effect of this polyphenol may require the presence of a viral envelope for its action [55]. At the same time, a pre-treatment of MDCK (Madin-Darby Canine Kidney cells) with HT did not affect the spread of H9N2 when subsequently inoculated; although, the H9N2 virus inactivated with hydroxytyrosol maintained its hemagglutinating activity unaltered and bound to MDCK cells in a similar way to untreated virus, implying that HT targets the virus and not the host cell [55].

Neuraminidase activity in the HT-treated virus also remained unchanged. However, in cells inoculated with HT-inactivated H9N2 virus, neither the protein nor the mRNA was detected. Electron microscopy analysis showed morphological abnormalities in the HT-treated H9N2 virus, with most of the structures found in the virions being atypical. These results suggest that the viral structure is altered by HT [55].

On the other hand, a very important current opportunistic infection in humans is acquired immunodeficiency syndrome (AIDS), caused by the immunodeficiency virus (HIV). Despite the existence of compounds and combinations against the HIV virus to reduce morbidity and mortality in patients, they do not cure the infection, and the necessary studies aimed at destroying this virus are being carried out.

In addition, anti-HIV therapies must be maintained over the long term with significant problems of drug resistance and chronic toxicity. They observed that olive leaf extract containing HT inhibits acute infection and cell-to-cell transmission of HIV-1 between uninfected MT2 cells cultured with HIV-1-infected Th9 cells at a dose-dependent concentration, with an IC_50_ of about 0.2 μg/mL [56]. In addition, this extract also inhibits HIV-1 replication as assayed by p24 expression in infected Th9 cells, with no cytotoxicity detected in uninfected target cells [57,58].

During the last five years, different papers have been published demonstrating the antiviral activity of polyphenols, proposing that many polyphenols are able to inhibit the replication of different types of SARS-CoV [59,60,61,62,63,64]. Several of these works revealed that polyphenols were able to inhibit HIV DNA polymerase α and reverse transcriptase and thus able to block DNA elongation by competing with incoming nucleoside triphosphates (NTPs) [60], as well as type 3C L-protease activity (3CL-Pro) [61,62,63].

Following this approach, Wu et al. [64] have demonstrated the antiviral capacity of polyphenols to significantly inhibit RNA-dependent RNA polymerase (RdRp) in SARS-CoV-2, with even greater inhibitory capacity than that found in one of the most effective inhibitors, the FDA-approved drug remdesivir, thus confirming the antiviral activity of some polyphenols.

## 3. Biological Activities of the Triterpenic Components of the Olive Tree

Pentacyclic triterpenes are a group of secondary plant metabolites that have important biological properties related to health and disease prevention. The main triterpenic components present in the olive tree are represented by compounds such as maslinic, oleanolic, ursolic, betulinic, tormentic, epipomolic, and euscamphenic acids, as well as the terpenic dialcohols, erythrodiol, and uvaol, all of them derived from lupeol, α-amyrin, and β-amyrin [65], being, of all of them, maslinic acid (MA) the most important both quantitatively and qualitatively. Maslinic acid, (2α,3β)-2,3-dihydroxyolean-12-en-28-oic acid, C_30_H_48_O_4_, presents a molecular mass of 472.72 (Figure 4).

Triterpenoids are molecules consisting of 30 carbon atoms grouped in five cycles of six carbon atoms with different substituents. This triterpene is synthesized via the cytosolic mevalonate pathway from acetyl-CoA which is converted to activated isoprene units. They are synthesized from the condensation of six molecules of activated isoprene. (3*S*)-2,3-epoxi-2,3-dihidroescualeno (OS) is a common precursor of all of them [65].

(3*S*)-2,3-epoxi-2,3-dihidroescualeno is a substrate for several OS cyclases or triterpene cyclases, including epoxi-2,3-dihidroescualeno-lanosterol cyclase. β-amyrin synthase (βAS) catalyzes the production of β-amyrin, which is the first pentacyclic triterpenoid; erythrodiol, oleanolic acid, and maslinic acid are then sequentially generated from β-amyrin [65].

The hydrophobic nature of the triterpene compounds present in olive extracts facilitates their incorporation into the target cells, probably in a similar way to molecules with similar structures such as the various steroid hormones, although, unlike these, the nature of the triterpene transporters is not yet known. Nevertheless, it is a fact that these compounds are introduced into cells to initiate their molecular mechanisms. In this regard, Peragón et al. [66] have demonstrated, by implementing a novel method based on the combination of high-performance liquid chromatography coupled to mass spectrometry (HPLC-MS/MS), the effective incorporation of maslinic acid and oleanolic acid into several cancer cell lines, HT29 and HepG2. In all cases, the transport system was dose dependent, although the dynamics of the entry behavior differed depending on the cell type [66]. The kinetics of entry of these compounds into HT29 cells was sigmoid, whereas in HepG2 cells it was linear, revealing specific dynamic mechanisms of entry into the cells [66].

A complete relationship of the molecular signaling pathways of the bioactivities of the main pentacyclic triterpenes present in the olive tree is shown in Figure 5. Table 2 summarizes the main cell lines and living organisms used in this review, together with the target cells and molecular markers in the bioactivity of the triterpenes present in olive extracts.

### 3.1. Stimulating Effects of Normal Growth

One of the main effects of different pentacyclic triterpenes is to act as true stimulators of cell and organ growth, mainly due to their ability to inhibit serine-protease activity [67,68] and to stimulate the production of reducing equivalents in the form of NADPH [69]. Both capacities were able to increase protein synthesis rates and reduce degradation rates in both trout liver and white muscle and liver and white-muscle sea bream [70,71,72,73].

In general, it is established that growth is a consequence of protein accumulation in different tissues, with muscle being the most representative tissue. The nature of growth can be interpreted in terms of an increase in the number or size of cells [74], although in target muscle tissue, where the possibility of polynucleated cells arises, changes in the total DNA and protein/DNA ratio cannot be so easily explained in the same terms, which is why the terms hypertrophy and hyperplasia can be used to describe changes in the size and number of DNA units [75].

In this sense, the administration of MA results in an increased total DNA content (hyperplasia) without changes in the protein/DNA ratio (hypertrophy) [67,68,70,71,72,73]. In summary, under these experimental conditions, MA stimulates the growth of the target muscle by inducing these processes of the generation of new muscle cells or fibers. The processes of hypertrophy of these cells predominate during the early stages of the animal’s growth, while in later stages of development, the generation of new muscle fibers is more important [75].

Moreover, the inclusion of MA in the animal diet significantly increased other growth-related parameters such as K_G_, protein accumulation rate, K_S_, protein synthesis rate, C_S_, protein-synthesis capacity, K_DNA_, protein synthesis rate/DNA unit, K_RNA_, protein-synthesis efficiency and PRE, and protein-retention efficiency, while maintaining the same or significantly reducing the K_D_, protein-degradation rate of the system [70,71,72,73]. As a consequence, MA-fed animals increased in terms of relative and absolute protein synthesis rate and protein-synthesis capacity, as well as absolute rates of protein accumulation. Therefore, these results suggest that MA promotes protein accumulation or protein reserve, which translates into increased body growth [70,71,72,73]. The main consequences of these behaviors were to show a significant increase in animal growth [70,72], together with a higher health index reflected in a significant decrease in animal mortality by significantly improving liver activity [71,72].

### 3.2. Immunomodulatory and Anti-Inflammatory Effects

Results from studies with MA in the catecholaminergic cells (CNS) showed that MA treatment exerts a potent anti-inflammatory action by inhibiting the production of nitric oxide and tumor necrosis factor-alpha (TNF-α) [76]. Furthermore, it significantly suppressed the expression of cyclooxygenase-2 (COX-2) and inducible nitric oxide synthase (iNOS) at protein and mRNA levels. In addition, this study showed how MA inhibits nitric oxide production and the secretion of inflammatory cytokines, IL-6 and TNFα, by macrophages [76]. Other studies indicate that MA can induce aortic vasodilation in hypertensive rats with an effect involving endothelial iNOS. In the case of maslinic acid, the activation of the protein kinase JNK would be closely linked to the immune-modulating capacity described by [77,78,79,80]. On the other hand, it is also indicated that the regulation of NF-κB by TNFα is affected by the action of this triterpene [77].

Pro-apoptotic activation of JNK is intimately linked to the inhibition of inflammatory and proliferative signals. This inhibits transcription factors such as NF-κB and the AP-1 cluster, which are responsible for the expression of inflammatory proteins such as COX-2 and iNOS, anti-apoptotic proteins of the Bcl-2 family, and angiogenesis and metastasis-related proteins such as ODC and matrix metallopeptidase 9 (MMP9), also known as type IV collagenase. In addition, MA can induce aortic vasodilation from hypertensive rats in an effect involving endothelial nitric oxide synthase (eNOS) [81].

In a recent study taking advantage of the anti-inflammatory capacity of triterpenes, Nagai et al., in an excellent clinical trial (CT) [82], have shown that MA intake was able to improve muscle response to resistance training through its anti-inflammatory capacity by reducing proinflammatory cytokines, previously shown by other authors, and that they are involved in atrophy of muscle fibers. After 12 weeks, skeletal and segmental muscle mass (right and left arm and trunk), as well as the knee pain score, improved significantly in the MA group, while these parameters, in the placebo group, were unchanged or, at worst, worsened. Therefore, these results indicate that MA could be effective in preventing mobility-related disability in the elderly [82].

### 3.3. Antioxidant and Anti-Ageing Effects

Cellular oxidative stress, recognized by changes in the levels of reactive oxygen species (ROS), is another fundamental aspect to understand the behavior and development of the carcinogenic process and that has a lot to do with the oxidative potential of the mitochondrial membrane (MMP). For this reason, the role of changes in the cellular concentration of molecules of reducing equivalents in the form of NADPH, both cytosolic and mitochondrial, is fundamental and, therefore, very important.

In fact, the molecules of this reduced coenzyme play a fundamental role in controlling ROS levels and in numerous metabolic processes involved in cell growth and development, serving as a connection between anabolic and catabolic processes [83,84], and are key in the redox balance [85,86,87,88], by actively participating in the detoxification processes [88], and during the maintenance of cellular integrity [86,87,88], as well as for their active role in cellular and organic aging [89].

The cellular levels of this reduced coenzyme depend on the enzymatic activity of the different NADPH production systems, especially those belonging to the pentose phosphate (PPP), glucose-6-phosphate dehydrogenase (G6PDH) and 6-phosphogluconate dehydrogenase (6PGDH), and also to NADP-dependent isocitrate dehydrogenase (NADP-ICDH). Especially important are its changes in concentration during the vital development of living beings due to its participation in the processes of cell growth [73,90,91,92], during cell differentiation [93,94,95] as well as in the maintenance of sensory and nutritional quality [89,91,92,96,97,98,99].

Many of these triterpenes have been shown to have significant antioxidant [100,101,102], anti-inflammatory [75], antimicrobial [103], antiviral [104,105,106], and even antitumor [68,77,78,79,107,108,109,110,111] effects; MA is considered to be one of the most potent triterpenes with a multitude of biological and therapeutic properties, including antioxidant, anti-inflammatory, antihypertensive, antiviral, antiangiogenic, antitumor, and antiangiogenic properties [77,78,79,107,108,109,110,111]. Montilla et al. [100], in a pioneering work, demonstrated the antioxidant effect of MA using rats previously treated with CCl_4_ and studied lipid peroxidation (LPO) of hepatocyte membranes induced ex vivo by the hydroxyl radicals generated by the Fe^2+^/H_2_O_2_ system and in vitro by the Fe^3+^/ascorbate system. Furthermore, these authors also showed that the endogenous plasma lipoperoxide (LP) levels and the susceptibility to LPO were significantly decreased in the presence of MA [100].

Several groups have described how MA decreases the susceptibility of hepatocyte membranes to lipid peroxidation, offering advantages in resistance to oxidative stress [100,112]. The antioxidant activity of MA was demonstrated in membranes of hepatocytes pre-treated with CCl_4_, where it was seen that MA produced a decrease in the levels of lipid peroxidation [101] and reduced H_2_O_2_ formation in macrophages treated with PMA (12-myristate-13-acetate-phorbol) [76]. Preliminary results obtained by our group show that MA is able to reduce intracellular ROS levels and modify the enzymatic defense system in B16F10 melanoma cells subjected to oxidative stress [85,102].

In addition, in a recent study, the antioxidant and antitumor capacity of erythrodiol (EO) was demonstrated in HepG2 liver carcinoma cells [113]. The results show that EO reduced the viability of HepG2 cells without changing ROS levels, whereas glutathione reduced (GSH) and NADPH levels were reduced, with selective changes in the activity of several antioxidant enzymes, catalase (CAT), superoxide dismutase (SOD), glutathione *S*-transferase (GST), glutathione peroxidase (GPX), glutathione reductase (GR), glucose 6-phosphate dehydrogenase (G6PDG), and 6-phosphogluconate dehydrogenase (6PGDH). Furthermore, EO was able to significantly modify the proteome of these cells, revealing that nuclear transport of mature mRNA was significantly reduced, while AMP biosynthesis and the G_2_/M phase transition of the cell cycle were induced, thus demonstrating its high anticancer potency [113].

### 3.4. Anti-Proliferative and Anticancer Effects

The anti-tumor activity of MA has become remarkable in recent years, as evidenced by the large number of studies addressing this issue [114], compared to those on other biological effects. The vast majority of published references correspond to in vitro experiments showing the antiproliferative and/or pro-apoptotic effect of MA [115], together with mechanisms of action involving different signaling pathways. Colon cancer cell lines have been widely used for this purpose and it has been shown that MA selectively induces cytotoxicity in human colon cancer cells, with lower IC_50_ values in HT29 cells (IC_50_: 32 µM) than in Caco-2 (IC_50_: 85 µM) after 72 h of incubation with the compound [76,77,78,107,108].

This effect has also been studied in other cell lines, finding different IC_50_ values depending on the cell lines and incubation time, for example, hepatocellular carcinoma HepG2 (IC_50_: 69.1 μM), murine melanoma B16-F10 (IC_50_: 36.2 μM), breast adenocarcinoma MCF-7 (IC_50_: 136.0 μM), salivary carcinoma ACC (IC_50_: 43.7 μM), urinary carcinoma T24 (IC_50_: 33.0 μM), urinary papilloma RT4 (IC_50_: 42.7 μM), synovial sarcoma SW982 (IC_50_: 45.3 μM), and leiomyosarcoma of uterus SK-UT 1 (IC_50_: 59.1 μM) [116,117,118].

Several studies have shown that MA exerts anti-proliferative and pro-apoptotic effects in HT29 and Caco-2, and the molecular mechanisms by which MA induces the apoptosis pathway have been described. Thus, in HT29, initially, MA activates the MAP kinase pathway, MAPKs, including JNK, and therefore, activates p53, causing cell cycle arrest and, at the same time, a delay in the induction of genotoxicity. Subsequently, in response to this activation of JNK and p53, the expression of the pro-apoptotic Bcl-2 family proteins Bax and Bid increases, while Bcl-2 expression decreases. The mitochondrial apoptotic pathway is activated, leading to mitochondrial disorganization and activation of caspase-9, which eventually leads to activation of caspase-3, caspase-8, and caspase-7 [77,78,79,107,108,119].

Furthermore, in Apc^Min/+^ mice, treatment with MA reduced the expression of the Rps6ka2 gene, which encodes the p90^Rsk^ protein, a mediator in the MAP kinase pathway [109], which in turn inhibits apoptosis induced through Bad activation. In colorectal cancer (CRC), it was shown that MA could activate MAPKs and negatively regulate the mTOR pathway [120]. Knockdown of AMPK abolished its inhibitory effect on cell proliferation and migration and blocked MA-induced apoptosis, revealing that AMPK was associated with the anticancer activity of MA. Furthermore, MA significantly suppressed tumorigenesis and up-regulated the AMPK/mTOR pathway in azoxymethane (AOM)/dextran sulphate sodium (DSS) and xenograft tumor mice [120].

Other effects of MA on the MAPK pathway have been described. Thus, the interaction of MA in the p38 MAPK cascade led to the induction of apoptosis in ACC2 and ACCM salivary carcinoma cells. MA resulted in the phosphorylation of p38 MAPK, as a consequence of increased intracellular Ca^2+^ levels [116]. A dose- and time-dependent increase in p38 phosphorylation during MA-induced apoptosis in urinary carcinoma T24 and 253J cells has also been described [118].

A recent study showed that betulinic acid (BA) is capable of inhibiting cancer progression in two colorectal cancer lines, HCT116 and SW480, by interfering with the levels of pro-apoptotic markers, increasing the levels of cleaved Bax and caspase-3 and caspase-9, and decreasing those of the antiapoptotic Bcl2. At the same time, this triterpene was able to modify the levels of the HSPA chaperone according to the concentrations of BA used by being able to bind to the nucleotide-binding domain of HSP70 in its ADP-binding state, indicating its high anticancer capacity [121].

Another triterpene from the olive tree, uvaol (UO), has been shown to be a potent anticarcinogenic agent. Bonel-Pérez et al. [86] have demonstrated this activity in liver carcinoma cells, HepG2, in comparison with a normal human liver cell line, WRL68, used as a control. Their results show significant anti-migratory activity together with significant pro-apoptotic activity by observing cell cycle arrest in the G_0_/G_1_ phase, an increase in the HSP60 chaperone and Bax, together with a decrease in Bcl2 and ROS. At the same time, these authors have demonstrated an anticancer role by modifying the AKT/PI3K-mediated signaling pathway [86].

At the end of 2001, the existence of small non-coding single-stranded RNA molecules with an average length of 22 nucleotides, called microRNAs (miRNA or miR), were described and published for the first time [122]. These molecules are capable of regulating the expression of their target genes, altering the repression of the encoded proteins through the repression of translation and/or the degradation of their mRNA [122]. They participate in practically all cellular processes, including metabolic, immunological, developmental and growth, proliferative, and survival [123,124,125]. In this context, in recent years, different studies have been published related to the expression of different types of miRNAs by some of the triterpenes mentioned in this review, in relation to their functions at metabolic and proliferative levels [126,127,128,129].

The treatment of two cancer cell lines (the triple-positive breast cancer BT474 and the triple-negative breast cancer MDA-MB-453) that overexpress ErbB2 (receptor tyrosine-protein kinase erbB-2) at different doses of betulinic acid (BA), showed, in a dose-dependent form, an inhibition of cell growth, an increase in cell apoptosis, and a reduction in the expression of specific protein (Sp) transcription factors, with the consequent decrease in ErbB2 expression due to the repression of the YY1 gene regulated by Sp [127,128]. The triggering of these molecular effects brought about a decrease in miR-27a, allowing an increase in ZBTB10 and a significant reduction in VEGFR (vascular endothelial growth factor receptor) levels, facilitating apoptosis and a significant decrease in angiogenesis [127,128].

The anticancer capacity of ursolic acid (UA) has been demonstrated, both in vitro and in vivo, in several cancer cell lines (the gastric cancer line, BGC-803, and the hepatocellular cancer H22 xenograft), activating both the pathway intrinsic apoptotic pathway as well as the extrinsic apoptotic pathway, modifying most of the molecular markers involved in them, such as cellular arrest in the G_0_/G_1_ phase, DNA fragmentation, an increase in the activities of caspases-3, caspase-8 and caspase-9, as well as downregulation of antiapoptotic proteins such as Bcl-2 [130,131]. Subsequently, these same authors have shown, in a human glioma cell line, U-251, that UA, in addition to triggering similar mechanisms of cytotoxicity, apoptosis, and survival, was capable of increasing caspase-3 activity and, at the same time, significantly repressing the expression of microRNA-21 (miR-21), which facilitates the expression of PDCD4 (programmed cell death protein 4), which is a pro-apoptotic protein encoded by the corresponding gene, suggesting that UA induces apoptosis in U251 cells via the TGF-b1/miR-21/PDCD4 pathway activated by UA [129].

### 3.5. Antiviral and Antimicrobial Effects

Several authors have also described both MA and oleanolic acid (OA) and several of their chemical derivatives as effective compounds with antiproliferative and antiviral capacity [132,133,134,135,136,137,138,139,140]; moreover, like oleanolic acid, MA and some of its different derivatives possess antibacterial activity [103] and a potent in vitro AIDS virus protease inhibitory activity [104,105,106,132,138,139,140]. In this sense, [104] have shown that MA has a potent inhibitory effect on HIV-1 protease activity and this effect was subsequently corroborated by other groups [105,138,139,140].

Similarly, studies by the same group [105] show the inhibitory activity of oleanolic and maslinic acids against human immunodeficiency virus replication. Similarly, five new triterpenes extracted from *Excoecaria acerifolia* were analyzed spectrophotometrically and showed moderate anti-HIV activity [140]. This activity has also been found to be more potent in this acid than in other triterpenes such as ursolic acid (UA), epipomolic acid, euscamphenic acid, and tormentic acid, and, at the same time, as potential agents in the fight against the human immunodeficiency virus type 1, HIV-1 [132,140].

A new coronavirus emerged at the end of December 2019, SARS-CoV-2, capable of generating a severe acute respiratory syndrome, generating a global pandemic known as coronavirus disease 2019 (COVID-19). From that moment on, numerous research groups were launched to achieve an effective vaccine, but at the same time, great efforts have been made to find compounds capable of effectively controlling the infection. Excellent candidates are different types of triterpenes as well as some of their chemical derivatives that have been shown to have high antiviral effectiveness by significantly affecting the protease activity of the virus, being especially effective a derivative of maslinic acid linked to an isoxazole chlorinated [141,142,143,144,145].

In summary, these works clearly demonstrate the antiviral effect of the different triterpenes, such as MA, BA, and UA, derived from the olive tree, as well as other terpenoids from other natural sources, by significantly inhibiting different SARS-CoV-2 proteases, such as major viral protease (3CL-Pro) with IC_50_ values between 3.2 and 14.4 µM, the PL-Pro (papain-like protease), the SGp-RBD (spike glycoprotein receptor binding domain), the RdRp (RNA polymerase RNA-dependent), and ACE2 (angiotensin-converting enzyme 2), all key to understanding viral infection. [141,144,145].

## 4. Discussion

If there is an organic-nutritional situation that can adequately justify the development of a state of full health, this is the ability to prevent, in the first place, and eliminate, in the second place, the diseases that may occur and afflict human beings. In this sense, nutrition plays a key role, so that if it contains compounds that can achieve this objective, an optimal healthy state could be significantly promoted.

The products derived from the fruit of the olive tree, the extra virgin oil, are key pieces of the health benefits of the so-called Mediterranean diet. Numerous international research groups have published a large number of works demonstrating the indisputable beneficial properties for health and especially those related to cardiovascular health, with a significant defense against a large number of neoplasms and with an ostensible improvement of the immune system. The main health benefits are due to the role of the phytochemicals present in the fruit and leaves of the olive tree (*Olea europaea*), such as polyphenols and pentacyclic triterpenes, and represented, mainly, by hydroxytyrosol and maslinic acid, respectively. These bioactive compounds, therefore, have applications in animal nutrition, both as functional supplements and nutraceuticals, as well as applications in cosmetics and pharmacy.

The olive tree is undoubtedly not the only plant source of the phytochemicals discussed in this study; flavonoids, polyphenols, and triterpenes, and many other plant sources possess them and would therefore have the same beneficial properties as those analyzed for the olive tree. Although many of the bibliographical citations shown in this study state this, it would be interesting to supplement this study with two recent studies [146,147].

The fruits of two species of the genus *Artocarpus*, endemic to the Asian continent, have traditionally been used both as a food source and for their therapeutic values. However, the characteristics and functional properties of their flowers have now been analyzed [146], identifying in the floral extracts a significant concentration of phenols and flavonoids with a high antioxidant capacity as they are able to modulate the gene expression of some of the molecular targets involved in this antioxidant response, indicating their potential use as supplements in human nutrition [146].

The therapeutic virtues of *Artemisia annua*, known as Chinese wormwood, have long been demonstrated and are associated with the presence of artemisinin, a sesquiterpene lactone, whose main activity lies in its high antimalarial capacity. The anti-inflammatory and anticarcinogenic effects of this compound and some of its secondary metabolites present in ethanolic extracts of this plant have recently been analyzed [147]. In this study, it was demonstrated that all extracts showed a protective action against the proinflammatory stimulus of LPS, with the leaf extract being more effective, which was also confirmed at the molecular level by repressing the gene expression of TNF-α factor in human neuroblastoma SH-SY5Y cells treated with LPS [147].

## 5. Concluding Remarks and Future Perspectives

All the activities that have been analyzed in this review, and especially those related to immunomodulatory, antioxidant, and anti-infectious capacities, allow us to conclude that the use of these phytochemicals in the diet would help to ostensibly improve the health state of any organism, facilitating animal welfare and therefore longevity. Consequently, and just as the basic components of the Mediterranean diet have a proven and significant effect on the health of humans, the inclusion of polyphenols and triterpenes from the olive tree in the animal diet can be considered an excellent choice to improve the healthy growth of animals.

A good example that summarizes everything that this review tries to expose is reflected in the work of Escobar et al. [148] in which the use, in diets of prediabetic obese rats, of nanotransporters, constructed with a polymer of polylactic acid-coglycolic acid and in which a set of natural triterpenic compounds are encapsulated, achieved, in addition to a greater gastrointestinal release of these triterpenes, a high efficacy in the reduction of obesity concomitant with diabetes together with significant changes in the main metabolic markers related to DMT2 (diabetes mellitus type 2). All this allows us to recognize that the inclusion of phytochemicals in the diet causes a significant increase in the health state of the animals.

Another interesting example of the use of these phytochemicals as dietary supplements or nutraceuticals is PB125, comprising extracts of three ingredients, with specific levels of carnosol/carnosic acid, withaferin A, and luteolin, which, because of their ability to activate the Nrf2 pathway, regulate cytoprotective genes and protect cells against oxidative stress, which is ideally suited to contribute effectively to healthy aging [23].

Finally, a clear fact, demonstrated by the history of humanity itself, is that infections by viruses, bacteria, or other organisms significantly shorten the longevity of living beings and especially those with previous pathologies. For all these reasons, it is essential and even vital that a “good” diet contains nutritional supplements that are capable of modulating the immune system to deal with these infections and/or that are directly capable of fighting against these external organisms. Some excellent examples that can be qualified as nutraceuticals are part of the phytochemicals, polyphenols, and triterpenes, discussed in this review.

## Figures and Tables

**Figure 1 molecules-27-02341-f001:**
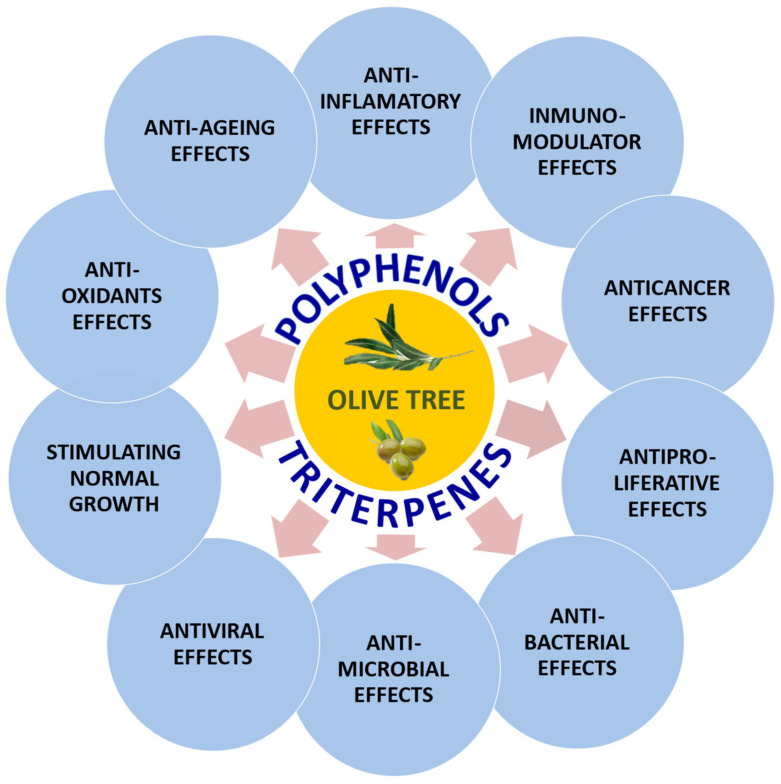
Main bioactivities of polyphenolic and triterpenoid compounds present in olive fruit and leaves related to the prevention and cure of different pathologies.

**Figure 2 molecules-27-02341-f002:**
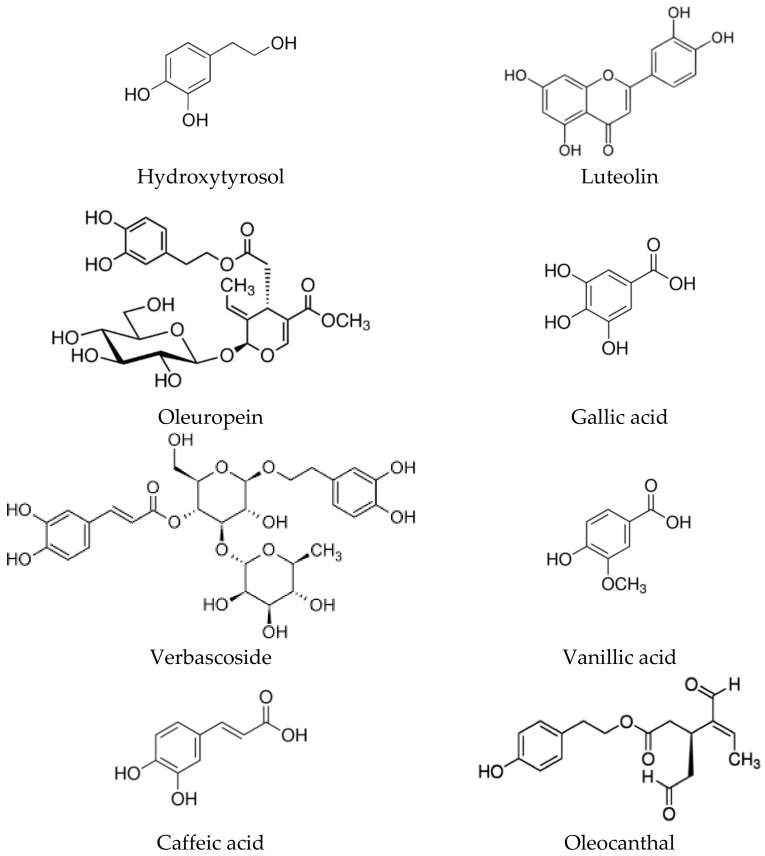
Chemical structure of the most important phenolic compounds (hydroxytyrosol, oleuropein, luteolin, verbascoside, aldehydes as, oleocanthal, and the acids as, gallic, vanillic, and caffeic) present in the extracts of olive tree.

**Figure 3 molecules-27-02341-f003:**
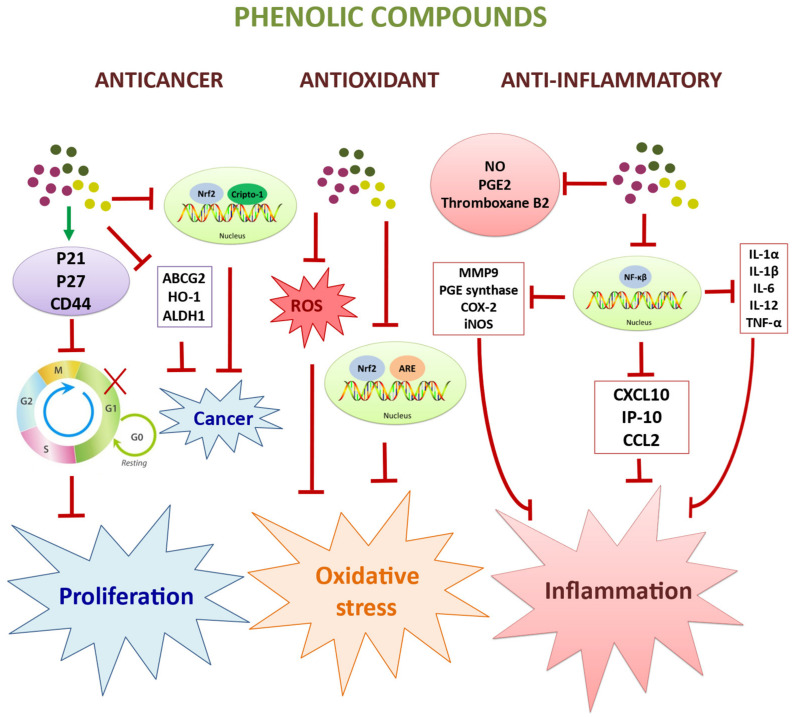
Schematic representation of the main molecular mechanisms involved in the action of the most important bioactivities related to the role of the polyphenols present in the fruit and leaf of the olive tree.

**Figure 4 molecules-27-02341-f004:**
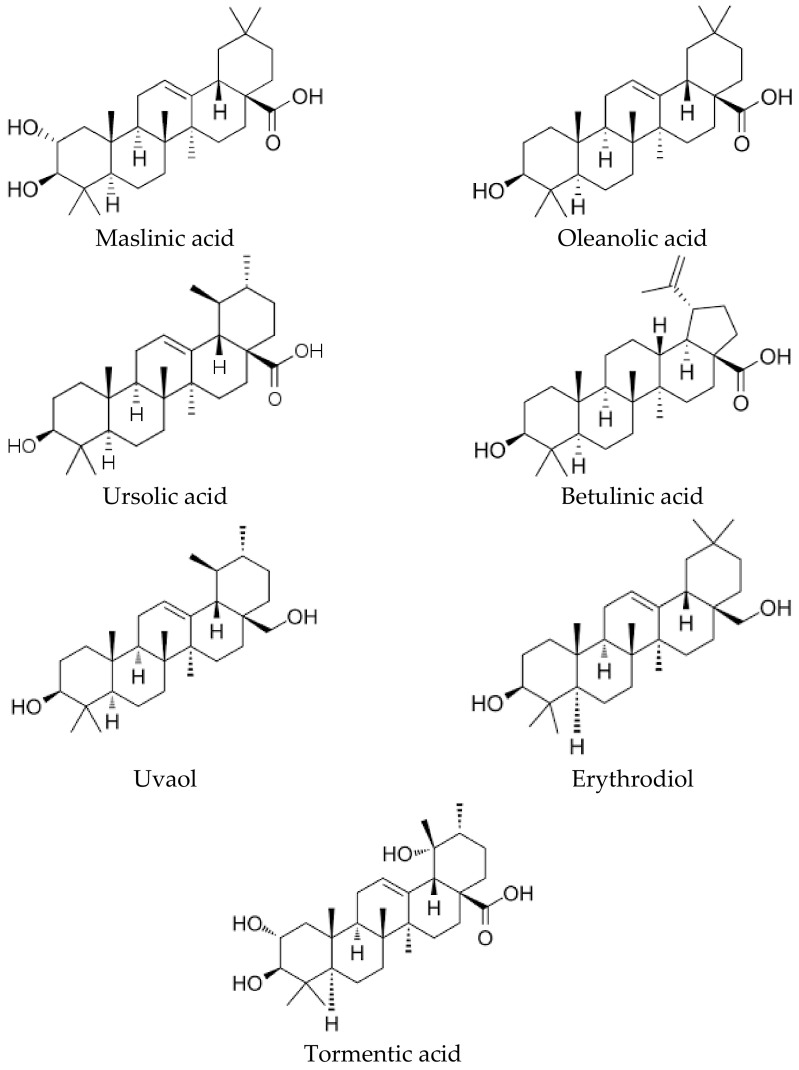
Chemical structure of the quantitatively most important triterpenoid compounds (acid and alcohols, maslinic, oleanolic, betulinic, ursolic, and tormentic acids and the triterpenic dialcohols, uvaol, and erythrodiol) present in the extracts of olive fruits and leaves.

**Figure 5 molecules-27-02341-f005:**
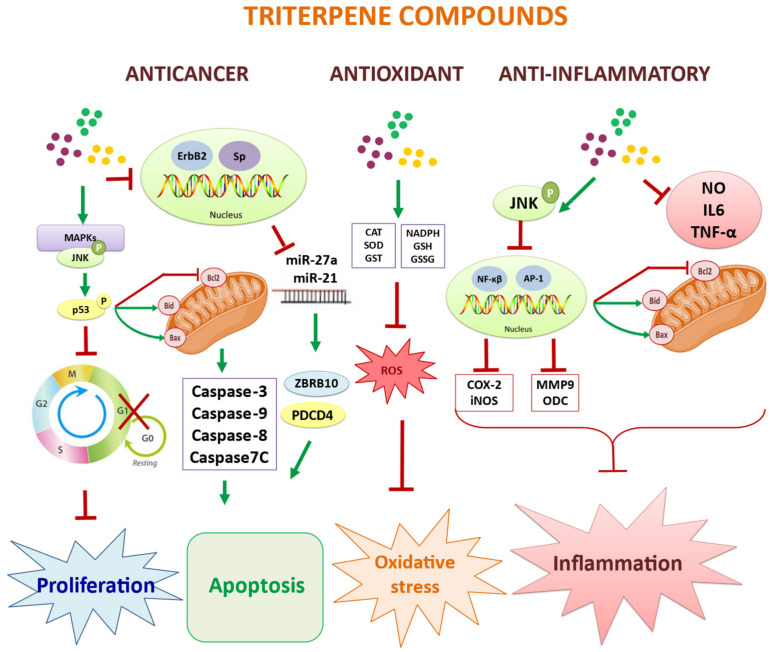
Schematic representation of the main molecular mechanisms involved in the action of the most important bioactivities related to the role of the triterpenoids present in the fruit and leaf of the olive tree.

**Table 1 molecules-27-02341-t001:** Main molecular markers that initiate in the bioactivities of the different polyphenols present in olive extracts and the target cells in which they are manifested.

Effects Parameters	Pharmaco-Kinetics and Toxicity [7,8,9,10,11,12,13,14,15,16,17,18,19]	Antioxidant and Anti-Aging Activity [20,21,22,23]	Anticancer and Antiproliferative Activity [24,25,26,27,28,29,30,31,32,33,34,35,36,37]	Immunomodulatory and Anti-Inflammatory [38,39,40,41,42,43,44,45,46,47,48,49]	Antiviral and Antimicrobial Activity [50,51,52,53,54,55,56,57,58,59,60,61,62,63,64]
Organisms, tissues, and cell lines studied in this review	Wistar rats Sprague-Dawley rats Caco-2 cells Human tissues Rat tissues	In vitro models Leucocyte cells HepG2 cells	HL60 cells HT29 cells MCF-7 cells MDA-MB-231 cells CSC cells HepG2 cells	RatsTHP-1 cells J774 cells Mouse macrophages	MT2 cells MDCK cells Th9 cells HIV virus SARS-CoV-2 Different types of bacteria
Molecular markers and target processes involved in the activity of olive’s fruit and leaf polyphenols	Bioavailability: Dose-dependent Rat tissues: Liver, Heart Spleen, Thymus Testicle, Brain Time absorption: 5–10 min after ingestion Recovery: 75% by aqueous solution 90% by oil solution Excretion: Glucuronide conjugates Toxicity: Nontoxic LD_50_ >>5 g/kg	Transcription factor: Nrf2 Redox homeostasis: ROS H_2_O_2_ O_2_^−^ LDL Glutathione NADPH Platelet Aggregation Chelating: Iron and copper metals Diseases: Cardiovascular Diabetes mellitus Metabolic syndrome Cancer Enzymes: GPX GR G6PDH 6PGDH	Cell cycle phases: G_0_/G_1_ G_2_/M S Cyclins: p21^WAF/Cip1^ p27^Kip1^ CD44 Pro-apoptotic Anti-metastatic Anti-angiogenic Enzymes and factors: Nrf2 ABCG2 HO-1 Cripto-1 ALDH1 Nuclear parameters: AMP biosynthesis Transport of mature mRNA	Parameters: NO PGE2 LPS Thromboxane B2 NF-κB Cytokines: IL-1α IL-1β IL-6 IL-12 TNF-α Chemokines: CXCL10/IP-10 CCL2/MPC-1 Enzymes: MMP9 PGE Synthase COX-2 iNOS	Viruses, bacteria: *Escherichia coli* *Candida albicans* *Kluyveromyces marxianus, Clostridium perfringens* *Streptococcus mutans, Shigella sonnei* *Salmonella enterica* *Vibrio* *parahaemolyticus, Vibrio cholerae,* *Salmonella typhi, Haemophilus* *influenzae,* *Staphylococcus aureus, Moraxella* *catarrhalis,* *Mycoplasma* *pneumoniae* Influenza A viruses: H1N1, H3N2, H5N1, H9N2, Newcastle virus SARS-CoV-2 DNA polymerase α, Reverse transcriptase, RdRp 3CL-Pro HIV virus p24 expression Viral proteases

Notes: The meaning of the bioactivity of olive polyphenols, represented by hydroxytyrosol, luteolin, oleuropein, gallic acid, verbascoside, vanillic acid, caffeic acid, oleocanthal, etc. Both increases and decreases in the different molecular markers are described in the text. The meaning of the abbreviations included in the table is defined in the abbreviations section at the end of the review.

**Table 2 molecules-27-02341-t002:** Main molecular markers that initiate in the bioactivities of the different pentacyclic triterpenes present in olive extracts and the target cells in which they are manifested.

Effects Parameters	Stimulating Effects of Normal Growth [67,68,69,70,71,72,73,74,75]	Immunomodulatory and Anti-Inflammatory [76,77,78,79,80,81,82]	Antioxidant and Anti-Aging Activity [83,84,85,86,87,88,89,90,91,92,93,94,95,96,97,98,99,100,101,102,103,104,105,106,107,108,109,110,111,112,113]	Anticancer and Antiproliferative Activity [114,115,116,117,118,119,120,121,122,123,124,125,126,127,128,129,130]	Antiviral and Antimicrobial Activity [131,132,133,134,135,136,137,138,139,140,141,142,143,144]
Organisms, tissues, and cell lines studied in this review	*Sparus aurata* *Oncorhynchus mykiss* Liver White muscle	Catecholaminergic cells Macrophages Aortic tissue	Hepatocyte Macrophage B16F10 cells Rats Plasma	HT29, Caco-2 cells HepG2 cells B16-F10 cells MCF-7, ACC cells T24, RT4 cells HCT116 cells SW982, 253J cells SK-UT-1 cells ACC2, ACCM cells WRL68 cells BT474, H22 cells BGC-803 cells	*Excoecaria acerifolia* sp. AIDS virus HIV-1 SARS-CoV-2
Molecular markers and target processes involved in the activity of olive’s fruit and leaf triterpenes	Hyperplasia Hypertrophy Liver White muscle NADPH Growth parameters: C_S_ K_D_ K_S_ K_G_ K_DNA_ K_RNA_ PRE Protein turnover: K_S_/K_D_	Molecular parameters: NF-κB AP-1 Bcl-2 family Bax, Bid Bcl2 Cytokines: IL-6 TNFα Enzymes: COX-2 iNOS eNOS JNK MMP9 or Collagenase type IV ODC Resistance training	Oxidative stress Parameters: ROS H_2_O_2_ NADP NADPH MMP CCl_4_ PMA GSH GSSG Lipid peroxidation (LPO) Enzymes as: GST CAT SOD G6PDH 6PGDH NADP-ICDH	Parameters: MAPKs, JNK activating p53 AMPK/mTOR Rps6ka2 gene p90^Rsk^ p38 MAPK AKT/PI3K ZBTB10, VEGFR ErbB2, Sp YY1, PDCD4 TGF-β1 Apoptotic proteins: Bcl2, Bax Bid, Bim Caspases: Caspase-3, -9 Caspase-8, -7 Chaperones: HSPA HSP70 HSP60 MicroRNAs miRNA-27a miRNA-21	Protease inhibitory activity: 3CL-Pro PL-Pro Antiviral capacity: SGp-RBD Virus replication: RdRp Viral repression: ACE2 Other triterpenic markers: Epipomolic acid Euscamphenic acid Tormentic acid Chemical derivatives: Maslinic isoxazole chlorinated

Notes: the meaning of the bioactivity of olive pentacyclic triterpenes, mainly represented by maslinic, oleanolic, ursolic, betulinic, tormentic acids, and the terpene dialcohols, uvaol, erythrodiol, etc. Both increases and decreases in the different molecular markers are described in the text. The meaning of the abbreviations included in the table is defined in the abbreviations section at the end of the review.

## Data Availability

Not applicable.

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
