# Peer review of "Nutraceutical Role of Polyphenols and Triterpenes Present in the Extracts of Fruits and Leaves of Olea europaea as Antioxidants, Anti-Infectives and Anticancer Agents on Healthy Growth"

_molecules, 2022, doi:10.3390/molecules27072341_

Round 1
Reviewer 1 Report
All of the parts I pointed out were corrected by the author.
Author Response
We again appreciate the excellent good review made by the reviewer as their insightful comments have made it possible for the current version of the manuscript to achieve a higher scientific quality.
Reviewer 2 Report
Reviewer Report of molecules-1654742-R1
Comments to Authors
The review articles describe "Nutraceutical Role of Polyphenols and Triterpenes Present in the Extracts of Fruits and Leaves of Olea europaea as Antioxidants, Anti-infectives and Anticancer Agents on Healthy Growth". Therefore, this review article aims to shed light on the nutraceutical role of two primary phytochemicals in Olea europaea fruit and leaf extracts, polyphenols and triterpenes onion healthy animal growth. Their immunomodulatory, anti-infective, antioxidant, anti-aging, and anti-carcinogenic capabilities show them to be potential nutraceuticals, providing healthy growth. Overall, some comments suggest the authors should be explained them well.
Comments from reviewer
- There were no references to support the data in Tables 1 and 2. In this review article, they seem extra pieces of information and do not have any relationship of goal in this manuscript. However, to avoid commercial activity, It was suggested to delete them.
- Figures 3 and 5, schematic representation of the main molecular mechanisms s involved in the action of the most important bioactivities related to the role of the polyphenols and triterpenoids present in the fruit and leaf of the olive tree. However, there were no following signaling pathways describing which were the target protein for polyphenols. It could not satisfactorily explain them and make many mistakes. Suggesting the authors must correct them and explain them well.
- In Figure3, the authors cannot support any pieces of evidence to support those polyphenols that could transport into the cell membrane. The authors only according to the described from Reference 10. However, that report only examines DPE intestinal transport's molecular mechanism, using differentiated Caco-2 cell monolayers as the model system. It could be passed through the gap junction Caco-2 cell monolayers. It could not mean those polyphenols can transfer from the cell membrane into plasma through the transporters.
- In Figure 5, the authors also cannot support any pieces of evidence to support those triterpenoids that could have activated the receptor on the cell membrane. Moreover, many of the antioxidative effects were relative to the mitochondria or some nuclear transcriptor proteins. However, the authors did describe them. Moreover, many inflammatory signal pathways began to form LPS binding the TLR co-receptors. However, the authors did not show the triterpenoids and TLR co-receptor relationship. Furthermore, the described anticancer pathways have many errors. Suggesting the authors must correct them and explain them well.
Author Response
General comments
First of all, we would like to sincerely thank the reviewer for all the review work done throughout these sessions. It is essential that genuine peer review is put in place if science is to move forward with soundness and quality. Secondly, the authors are grateful for the quality and scientific honesty of the reviewer in all the questions, suggestions and criticisms made with the aim of improving the quality of the manuscript. Thank you again.
Specific comments
Response 1
The reviewer's comments are very true, although our initial intention in including both tables was to show future readers the possibility of knowing about the existence of certain types of extracts enriched in different phytochemicals in order to recognize their nutraceutical nature. However, on this second occasion, we agreed with the reviewer that, in order to avoid confusion, possible errors and ambiguities about possible conflicts of interest, we decided, on his advice, to remove them from the text, knowing that the main composition of these phytochemicals is perfectly reflected throughout the text of the manuscript.
Response 2
Thank you again for your comment. The content of figures 3 and 5 have been from the beginning of their conception to reflect in a very summarized way all or most of the molecular effects that have been described for polyphenols and triterpenes. In these two figures, we have only indicated the most significant ones, having been described by the different authors who have published their work. No unpublished mechanisms have been introduced. A complete overview of all the mechanisms is given throughout the text. If there are mechanistic errors, these will be those of the published articles.
Response 3
The reviewer's point is very true as there is not a great deal of evidence for the transport of these phytochemicals across the plasma membrane. In the reference mentioned by the reviewer in which a specific cell type is used, it seems to be a bidirectional passive diffusion transport and is described as such in the manuscript. However, one fact is certain, and that is that these polyphenols or metabolic intermediates, generated during their cellular metabolism, are found inside cells to fulfil their various bioactivities.
In order to clarify this point, we have introduced several paragraphs that help to clarify and understand this problem, which I transfer below.
“Different authors (13) studied the tissue distribution of intravenously administered 14C-labeled HT in rats. At 5 minutes after injection, less than 8% was still present in the blood, approximately 6% in the plasma and 1.9% in the cellular fraction of the blood, so it was estimated that the half-life of the compound in blood was very small, below 2 minutes. Similar levels of 14C were found in the cells of skeletal muscle, liver, lungs, and heart, while the kidney cells accumulated 10 times more than the other organs. 14C was also detected in the brain, indicating that HT was able to cross the blood-brain barrier, although it has been described that HT can also be generated endogenously in the brain from dopamine (14) and from dihydroxyphenylacetic acid (DHPA) via dihydroxyphenylacetic reductase present in the brain (15).
HT metabolites were found in blood 5 minutes after intravenous injection of 14C-HT, indicating that the compound was rapidly metabolized in cells, especially in the liver and enterocytes, proposing three metabolic pathways for HT: Oxidation, through the enzymes alcohol-dehydrogenase and aldehyde-dehydrogenase giving rise to dihydroxyphenylacetic acid; methylation, through the enzyme catechol-O-methyltransferase (COMT) giving rise to homovanillic alcohol (HVOL); and methylation plus oxidation, to form homovanillic acid (HVA) (13).
Regarding excretion, it was found that 90% of the radioactivity was detected in the urine 5 hours after the intravenous injection of 14C-HT and a small proportion was excreted in the feces. (13), These results coincide with the results obtained in humans where most of the HT and tyrosol were found in the urine collected during the first 4 hours after administration. ingestion of 50 mL of virgin olive oil (16)."
In addition, in order to avoid possible errors, confusion and ambiguities, we have significantly modified figure 3 in the hope that all the reviewer's doubts will be clarified.
Response 4
In this case the reviewer is again correct. An error in the preparation of the figure, which undoubtedly was not reviewed by us, the authors of the manuscript, caused us to use a membrane receptor for triterpenes, knowing that these compounds, due to their hydrophobic nature, can cross the membrane through specific receptors, although, unlike structurally similar compounds such as steroid hormones, they have not been described, or at least we are not aware of them. Therefore, and in agreement with the reviewer, we have removed that part of figure 5 and have also introduced a new paragraph alluding to this phenomenon, in order to clarify this point, which I transfer below:
“The hydrophobic nature of the triterpene compounds present in olive extracts facilitates their incorporation into the target cells, probably in a similar way to molecules with similar structures such as the various steroid hormones, although, unlike these, the nature of the triterpene transporters is not yet known. Nevertheless, it is a fact that these compounds are introduced into cells to initiate their molecular mechanisms. In this regard, Peragón et al. (66) have demonstrated, by implementing a novel method based on the combination of high-performance liquid chromatography coupled to mass spectrometry (HPLC-MS/MS), an effective incorporation of maslinic acid and oleanolic acid into several cancer cell lines, HT29 and HepG2. In all cases, the transport system was dose-dependent, although the dynamics of the entry behavior differed depending on the cell type (66). The kinetics of entry of these compounds into HT29 cells was sigmoid, whereas in HepG2 cells it was linear, revealing specific dynamic mechanisms of entry into the cells (66)."
Reviewer 3 Report
The authors propose an in-depth review of the literature on the uses of derivatives extracted from the fruits and leaves of Olea europea.
I had already revised this manuscript and had suggested small revisions to improve the text and organization of the Review. The authors followed my advice and integrated the missing parts. Furthermore, they discussed in more detail some application aspects of Olea europea extracts.
Now, the manuscript has improved in every part. I suggest publication without further revision.
Author Response
All the authors of this review are deeply grateful for the work carried out by the reviewer, as his comments, criticisms and suggestions have made it possible to improve the article significantly and, at the same time, allow this necessary peer review to allow science to advance with security and solidity.
Reviewer 4 Report
The Authors corrected all the points I previously objected. Therefore, the acceptance of the current and revised version is suggested.
Author Response
First of all we want to deeply and sincerely thank the suggestions, criticisms and comments made by the reviewer during the two review sessions. His clever and brilliant suggestions have allowed the manuscript to finally reach sufficient academic quality to be accepted. Thanks from all the authors.
Round 2
Reviewer 2 Report
Reviewer Report of molecules-1654742-R2
Comments to Authors
The review articles describe "Nutraceutical Role of Polyphenols and Triterpenes Present in the Extracts of Fruits and Leaves of Olea europaea as Antioxidants, Anti-infectives and Anticancer Agents on Healthy Growth". Therefore, this review article aims to shed light on the nutraceutical role of two primary phytochemicals in Olea europaea fruit and leaf extracts, polyphenols and triterpenes onion healthy animal growth. Their immunomodulatory, anti-infective, antioxidant, anti-aging, and anti-carcinogenic capabilities show them to be potential nutraceuticals, providing healthy growth. Overall, some figures were not cited references well and still have a lot of errors in this manuscript. Some comments suggest the authors should be explained them well.
Comments from reviewer
- There were no references to support the data in Tables 1 and 2. In this review article, they seem extra pieces of information and do not have any relationship to the goal in this manuscript. However, to avoid commercial activity, It was suggested to delete them.
Response 1
The reviewer's comments are very true, although our initial intention in including both tables was to show future readers the possibility of knowing about the existence of certain types of extracts enriched in different phytochemicals in order to recognize their nutraceutical nature. However, on this second occasion, we agreed with the reviewer that, in order to avoid confusion, possible errors and ambiguities about possible conflicts of interest, we decided, on his advice, to remove them from the text, knowing that the main composition of these phytochemicals is perfectly reflected throughout the text of the manuscript.
Comment R2-1 from the reviewer
No more extra comments.
- Figures 3 and 5, are schematic representations of the main molecular mechanisms are involved in the action of the most important bioactivities related to the role of the polyphenols and triterpenoids present in the fruit and leaf of the olive tree. However, there were no following signaling pathways describing which were the target protein for polyphenols. It could not satisfactorily explain them and made many mistakes. Suggesting the authors must correct them and explain them well.
Response 2
Thank you again for your comment. The content of figures 3 and 5 have been from the beginning of their conception to reflect in a very summarized way all or most of the molecular effects that have been described for polyphenols and triterpenes. In these two figures, we have only indicated the most significant ones, having been described by the different authors who have published their work. No unpublished mechanisms have been introduced. A complete overview of all the mechanisms is given throughout the text. If there are mechanistic errors, these will be those of the published articles.
Comment R2-2 from the reviewer
It was well known that only tyrosine could transport into the noradrenergic nerve ending or varicosity by a sodium-dependent carrier (https://www.ncbi.nlm.nih.gov/books/NBK540977/), then tyrosine is converted to dopamine (double hydroxy group). There were no signaling pathways describing which were the target protein for polyphenols and could transport them into the cellular membrane. However, the reviewer disagrees with the opinion “If there are mechanistic errors, these will be those of the published articles”. Overall, the mistake could come from the authors did not fully understand those mechanisms.
- In Figure3, the authors cannot support any pieces of evidence to support those polyphenols that could transport into the cell membrane. The authors only according to the described in Reference 10. However, that report only examines DPE intestinal transport's molecular mechanism, using differentiated Caco-2 cell monolayers as the model system. It could be passed through the gap junction Caco-2 cell monolayers. It could not mean those polyphenols can transfer from the cell membrane into plasma through the transporters.
Response 3
The reviewer's point is very true as there is not a great deal of evidence for the transport of these phytochemicals across the plasma membrane. In the reference mentioned by the reviewer in which a specific cell type is used, it seems to be a bidirectional passive diffusion transport and is described as such in the manuscript. However, one fact is certain, and that is that these polyphenols or metabolic intermediates, generated during their cellular metabolism, are found inside cells to fulfil their various bioactivities.
In order to clarify this point, we have introduced several paragraphs that help to clarify and understand this problem, which I transfer below.
“Different authors (13) studied the tissue distribution of intravenously administered 14C-labeled HT in rats. At 5 minutes after injection, less than 8% was still present in the blood, approximately 6% in the plasma and 1.9% in the cellular fraction of the blood, so it was estimated that the half-life of the compound in blood was very small, below 2 minutes. Similar levels of 14C were found in the cells of skeletal muscle, liver, lungs, and heart, while the kidney cells accumulated 10 times more than the other organs. 14C was also detected in the brain, indicating that HT was able to cross the blood-brain barrier, although it has been described that HT can also be generated endogenously in the brain from dopamine (14) and from dihydroxyphenylacetic acid (DHPA) via dihydroxyphenylacetic reductase present in the brain (15).
HT metabolites were found in blood 5 minutes after intravenous injection of 14C-HT, indicating that the compound was rapidly metabolized in cells, especially in the liver and enterocytes, proposing three metabolic pathways for HT: Oxidation, through the enzymes alcohol-dehydrogenase and aldehyde-dehydrogenase giving rise to dihydroxyphenylacetic acid; methylation, through the enzyme catechol-O-methyltransferase (COMT) giving rise to homovanillic alcohol (HVOL); and methylation plus oxidation, to form homovanillic acid (HVA) (13).
Regarding excretion, it was found that 90% of the radioactivity was detected in the urine 5 hours after the intravenous injection of 14C-HT and a small proportion was excreted in the feces. (13), These results coincide with the results obtained in humans where most of the HT and tyrosol were found in the urine collected during the first 4 hours after administration. ingestion of 50 mL of virgin olive oil (16)."
In addition, in order to avoid possible errors, confusion and ambiguities, we have significantly modified figure 3 in the hope that all the reviewer's doubts will be clarified.
Comment R2-3 from the reviewer
Figure 3. Schematic representation of the main molecular mechanisms involved in the action of the most important bioactivities related to the role of the polyphenols present in the fruit and leaf of the olive tree. This figure was not cited references well and still has a lot of errors. However, those phenolic compounds do not belong to the molecular mechanisms in the signaling pathways. These phenolic compounds just only interfere with some functional proteins (like receptors, ion channels, phosphatase, kinase, secondary message proteins…). Figure 3 still has a lot of errors and did not cite references well. Suggesting the authors must correct those errors in Figure 3 and explain them well in detail.
4. In Figure 5, the authors also cannot support any pieces of evidence to support those triterpenoids that could have activated the receptor on the cell membrane. Moreover, many of the antioxidative effects were relative to the mitochondria or some nuclear transcriptor proteins. However, the authors did describe them. Moreover, many inflammatory signal pathways began to form LPS binding to the TLR co-receptors. However, the authors did not show the triterpenoids and TLR co-receptor relationship. Furthermore, the described anticancer pathways have many errors. Suggesting the authors must correct them and explain them well.
Response 4
In this case the reviewer is again correct. An error in the preparation of the figure, which undoubtedly was not reviewed by us, the authors of the manuscript, caused us to use a membrane receptor for triterpenes, knowing that these compounds, due to their hydrophobic nature, can cross the membrane through specific receptors, although, unlike structurally similar compounds such as steroid hormones, they have not been described, or at least we are not aware of them. Therefore, and in agreement with the reviewer, we have removed that part of figure 5 and have also introduced a new paragraph alluding to this phenomenon, in order to clarify this point, which I transfer below:
“The hydrophobic nature of the triterpene compounds present in olive extracts facilitates their incorporation into the target cells, probably in a similar way to molecules with similar structures such as the various steroid hormones, although, unlike these, the nature of the triterpene transporters is not yet known. Nevertheless, it is a fact that these compounds are introduced into cells to initiate their molecular mechanisms. In this regard, Peragón et al. (66) have demonstrated, by implementing a novel method based on the combination of high-performance liquid chromatography coupled to mass spectrometry (HPLC-MS/MS), an effective incorporation of maslinic acid and oleanolic acid into several cancer cell lines, HT29 and HepG2. In all cases, the transport system was dose-dependent, although the dynamics of the entry behavior differed depending on the cell type (66). The kinetics of entry of these compounds into HT29 cells was sigmoid, whereas in HepG2 cells it was linear, revealing specific dynamic mechanisms of entry into the cells (66)."
Comment R2-4 from the reviewer
Figure 5. Schematic representation of the main molecular mechanisms involved in the action of the most important bioactivities related to the role of the triterpenoids present in the fruit and leaf of the olive tree. However, those triterpenoids do not belong to the molecular mechanisms in the signaling pathways. Moreover, some hydrophobic characteristics of the triterpenoids could be transported into the target cells directly and interfere with some functional proteins (like receptors, ion channels, phosphatase, kinase, secondary message proteins…). Figure 5 still has a lot of errors and did not cite references well. Suggesting the authors must correct those errors in Figure 5 and explain them well in detail.

This manuscript is a resubmission of an earlier submission. The following is a list of the peer review reports and author responses from that submission.
Round 1
Reviewer 1 Report
General comments:
The author described the anti-cancer, antioxidant, antiviral properties of hydroxytyrosol and maslinic acid, which are one of the phenols and terpenoids found in olive trees. The work presented for review is valuable, however I’d like to request that the author make a few changes to make it more understandable to the readers.
Specific points:
Title: The word "olive" is strangely placed in the title-> olive fruit and leaf?? Or fruit and leaf of olea europaea?
Abstract: It appears that the abstract does not accurately reflect the content of the review. The first four sentences could be removed or relocated to the introduction and the composition of the olive tree phytochemicals and their functions should be discussed more in the abstract.
Figure 1: Although the roles of polyphenols and triterpenes are nearly identical, why not exhibit them together in a Venn diagram? And, if the cardioprotector effect isn't mentioned in the review, why not leave it out?
Table 3 and 4: Because the parameter title is odd, why don't you summarize them to “Cell lines and organisms used in this review” and “ Target cells and molecular markers involved” ?
Figure 3 and 4: The OE, CA, UO, and EO markings on the figures, in my opinion, should be corrected. These chemicals have not been reviewed to have all those functions.
Please review your manuscript for TYPOS and grammatical errors.
66: Erase period
119: Is-> It is
245: antiproliferaty effects-?
Author Response
REVIEWER 1
Article: molecules-1593262
Title: Role of Phytochemicals (Polyphenols and Triterpenes) present in Fruit and Olive Leaf Extracts as Antioxidants, Anti-infectives and Anticarcinogens in the Healthy Animal Growth
Authors: Rufino-Palomares, E.E. et al.
General comments:
First of all, the authors of the present review would like to deeply thank the different reviewers of the paper, for the time dedicated to the review of the paper for two fundamental reasons; the first reason, because of the absolute necessity that all scientific papers submitted for publication must be endorsed and accepted after peer review, as that is the only way to ensure their scientific potential and thus be able to advance, in an adequate and safe way, science. The second reason is that the criticisms made by peer review can significantly improve the quality of the work itself.
Secondly, we would also like to thank the different reviewers for both the good criticisms made of our work and the less favourable ones, as we understand that both types of criticisms undoubtedly help to increase the quality of the work. We believe that all the comments are worth taking into account and we are going to reply to them point by point and to modify and/or qualify everything that we believe may serve to improve the review and are in accordance with the objectives set out in the review.
RESPONSES TO REVIEWER 1
We thank the reviewer for the good critique of our work and we think that all his comments will improve the understanding of future readers.
1.- Title
It is quite true that the word olive as we have presented it in the text is placed in a strange way and therefore, we are going to modify it according to his indications, and it will be as follows:
"Nutraceutical Role of Polyphenols and Triterpenes Present in the Extracts of Fruits and Leaves of Olea europaea as Antioxidants, Anti-infectives and Anticancer Agents on Healthy Growth"
2.- Abstract
Although at first, and due to its importance, we thought of using the section of the abstract to justify the indisputable fact that proper nutrition is a key element in understanding its role in animal health, by linking it to the benefits of a Mediterranean diet, based on the intake of nutrients such as virgin olive oil; once we reread it, we agree with the reviewer and rewrite the abstract in accordance with his indications, relocating this justification to the Introduction section.
3.- Figure 1
Figure 1 shows the bioactive effects of the two main components present in olive fruit and leaf extracts and although they are practically similar, differing in their greater or lesser specificity, we believe that by repeating them in the Venn diagram it is possible to recognise them separately and intuit the synergy that can occur when they are ingested together. On the other hand, it is true that other effects appear which have not been dealt with in the review, and although they are part of their bioactivities, we believe, like the reviewer, that they can be eliminated from them, following his indications.
4.- Tables 3 and 4
In tables 3 and 4, we have sought to reflect two different aspects of our study. On the one hand, to indicate the organic models, whether they are whole organisms, tissues, primary cell cultures or different cell lines, used by the authors of the studies that have served as the basis of this review, and on the other hand, the molecular effects that these components cause on the target cells used, including the main molecular elements involved in the signalling pathways that they trigger. Notwithstanding this, we have modified the tables to some extent, following some of the reviewer's indications, although we believe it is necessary to keep the two aspects separate.
5.- Figures 3 and 4
We believe that the reviewer is referring to figures 3 and 5. We wanted these figures to reflect a summary of all the intracellular signalling pathways involved in the effects of the different polyphenols and triterpenes present in olive extracts and never that all these components cause all these effects. Therefore, and this has been our intention, that the figures reflect all of these pathways and never that they are caused by each and every one of these components. To avoid this possible confusion, the initials of the components have been removed from the figures.
6.- Typographical and grammatical errors
It is true that this type of error always occurs and we thank the reviewer for this comment. We have thoroughly revised the entire manuscript and corrected all those we have detected.
With regard to the errors you mention, we have not found the one referring to line 66.
We have corrected the one on line 119.
The comment/question corresponding to line 245, we wanted to include the antiproliferative effects (fundamentally cytotoxicity) which, although they are apparently the same or similar to that of anticancer (fundamentally apoptosis, angiogenesis, etc.), there are important differences and this is reflected in some of the studies used in this review.
We thank the Reviewer for all comments and insightful suggestions.
Reviewer 2 Report
Reviewer Report of molecules-1593262
Comments to Authors
The review articles describe "Role of Phytochemicals (Polyphenols and Triterpenes) present in Fruit and Olive Leaf Extracts as Antioxidants, Anti-infectives, and Anticarcinogens in the Healthy Animal Growth". As shown in the manuscript, the main objective of this review article wants to shed light on the role of polyphenols and triterpenes in healthy animal growth. Overall, some comments suggest the authors should be explained them well.
Comments
- The review article was very similar to the other articles related to the fruit and olive leaf extracts but still did not have any novel viewpoints.
- The title and the abstract did not coordinate and reflex the manuscript's content.
- The review article included OLIVEMAS® and OLIVESAN® but did explain any essential pieces of information. However, they are trademark products in the markets. It seems to have some business behavior.
- Figures 2 and 3 have inaccurate descriptions, and many signaling pathways were wrong and lacked enough pieces of information to support them.
Author Response
REVIEWER 2
Article: molecules-1593262
Title: Role of Phytochemicals (Polyphenols and Triterpenes) present in Fruit and Olive Leaf Extracts as Antioxidants, Anti-infectives and Anticarcinogens in the Healthy Animal Growth
Authors: Rufino-Palomares, E.E. et al.
General comments:
First of all, the authors of the present review would like to deeply thank the different reviewers of the paper, for the time dedicated to the review of the paper for two fundamental reasons; the first reason, because of the absolute necessity that all scientific papers submitted for publication must be endorsed and accepted after peer review, as that is the only way to ensure their scientific potential and thus be able to advance, in an adequate and safe way, science. The second reason is that the criticisms made by peer review can significantly improve the quality of the work itself.
Secondly, we would also like to thank the different reviewers for both the good criticisms made of our work and the less favourable ones, as we understand that both types of criticisms undoubtedly help to increase the quality of the work. We believe that all the comments are worth taking into account and we are going to reply to them point by point and to modify and/or qualify everything that we believe may serve to improve the review and are in accordance with the objectives set out in the review.
RESPONSES TO REVIEWER 2
We are sorry that some aspects of our work do not seem right to the reviewer, although we are grateful for his comments and questions, which we will try to answer and modify accordingly.
Response 1
It is true that, throughout the existing bibliography in the scientific area, different reviews of the same subject are published and that apparently, they have some similarities, although a deeper reading of them allows us to recognise really different aspects. In this case, we have to say that there have not been many reviews so far related to the phytochemical components of the olive tree and their relationship with the nutritional benefits they provide for healthy animal growth. Our own group published a review on the anti-cancer and anti-angiogenic role of the pentacyclic triterpenes present in the plant world and since then there have been many new studies published that have provided new insights into the bioactive role of these components.
In contrast to the reviewer's comments, this review provides new data on the immunomodulatory, anticarcinogenic, antiproliferative, anti-inflammatory, antioxidant, antibacterial and antiviral role and the molecular mechanisms triggered by the action of these nutrients. As an example of novelty, among many others that are included in this manuscript, we would like to explain to the reviewer the inclusion of the role of the micro RNAs of some of these components in these molecular mechanisms related to many of the pathologies described and that negatively affect healthy animal growth.
Furthermore, a fact that shows the novelty of a significant part of the latest published works analysed in this study is the fact that more than 25% of the bibliographic citations in this study correspond to the last five years and that more than 5% correspond to the year 2021.
Response 2
The reviewer is right in his comment, as there seems to be no apparent link between what the title of the paper suggests and the content of the abstract. We originally wanted to use the section of the Abstract to emphasise that nutrition is an essential pillar of animal growth, and one way of dealing with this essential point is to use this section to justify it, a justification based on the behaviour of living things as open systems and what this means for their own development.
However, and according to the reviewer's comment we have changed the content of the abstract according to his suggestion and the new text is now in the revised manuscript.
Response 3
The fact that this review deals with the nutraceutical role of the components of olive tree extracts on healthy growth has motivated us to include two representative tables of two types of enriched extracts, one in polyphenols and the other in triterpenes, which could serve as a nutritional supplement. In our geographical area there is a company dedicated to the production of different types of natural extracts used in nutrition. The company is Extractos y Derivados, S.L. (Granada, Spain) as described at the bottom of both tables.
In agreement with the reviewer and in order not to interfere with any kind of product branding or commercial competition we can remove the trademarks from the title of both tables.
Response 4
The reviewer is most certainly referring to figures 3 and 5 where we illustrate the main intracellular signalling mechanisms triggered by the different components of olive extracts, polyphenols and triterpenes. It is not true that the mechanisms described in the figures are not properly contrasted in the information implicit in the manuscript. Precisely, not all of them are described in the text, but the most significant ones are.
These figures do not attempt to report the individualised mechanisms of each of the polyphenolic or triterpene components analysed in this review; they are, in fact, a global and schematic summary of most of the signalling pathways set in motion by these components as a whole, and what is certain is that all of them are duly described throughout the text of the manuscript.
Reviewer 3 Report
The authors present an overview of the biological activity of some components deriving from the fruits and leaves of Olea europaea.
The topic was developed in an extensive and in-depth manner. The text is clear and has an updated literature. Furthermore, the authors have enriched the text with figures, tables and diagrams that enrich the value of the manuscript.
However, in order to make the manuscript more original than many others already in the literature, I have some suggestions for the authors:
All the biological activities reported by the authors are mainly associated with molecules that can also be found in other plant species. What makes the difference in biological activity is the titration of the active components.
It would be interesting to report a paragraph on the different analytical approaches used to titrate the active ingredients present in the derivatives of O. europaea. For example see what has been done for other plant species:
Gupta, A.K.; et al. Artocarpus lakoocha Roxb. and Artocarpus heterophyllus Lam. Flowers: New Sources of Bioactive Compounds. Plants 2020, 9, 1329. doi: 10.3390/plants9101329
Abate, G. et al.. Phytochemical Analysis and Anti-Inflammatory Activity of Different Ethanolic Phyto-Extracts of Artemisia annua L.. Biomolecules 2021, 11, 975. doi: 10.3390/biom11070975
Another aspect that would enhance the manuscript is a geographical approach. Indeed, the growth area of Olea europaea can modify the content of secondary metabolites? Do the nutritional values change depending on the O. europaea derivatives collection area? These aspects should be discussed.
Author Response
REVIEWER 3
Article: molecules-1593262
Title: Role of Phytochemicals (Polyphenols and Triterpenes) present in Fruit and Olive Leaf Extracts as Antioxidants, Anti-infectives and Anticarcinogens in the Healthy Animal Growth
Authors: Rufino-Palomares, E.E. et al.
General comments:
First of all, the authors of the present review would like to deeply thank the different reviewers of the paper, for the time dedicated to the review of the paper for two fundamental reasons; the first reason, because of the absolute necessity that all scientific papers submitted for publication must be endorsed and accepted after peer review, as that is the only way to ensure their scientific potential and thus be able to advance, in an adequate and safe way, science. The second reason is that the criticisms made by peer review can significantly improve the quality of the work itself.
Secondly, we would also like to thank the different reviewers for both the good criticisms made of our work and the less favourable ones, as we understand that both types of criticisms undoubtedly help to increase the quality of the work. We believe that all the comments are worth taking into account and we are going to reply to them point by point and to modify and/or qualify everything that we believe may serve to improve the review and are in accordance with the objectives set out in the review.
RESPONSES TO REVIEWER 3
We would also like to thank this reviewer for his good critique of our study, as well as for his suggestions for improving our manuscript.
1.
In this regard, he made two suggestions. The first is the need to report on the different analytical approaches used to assess the active principles present in Olea europaea derivatives. Specifically in this regard, we would like to comment that our research group has been working for the last 25 years in the field of bioactivities of natural components and very closely with research groups of the Department of Organic and Analytical Chemistry of our University with the capacity to identify, title and synthesise a large number of active ingredients, Due to this close collaboration, we are currently writing, jointly with both research groups, a review on the subject proposed to us, which we believe will be of great interest to specialists in the field. However, due to this interesting suggestion, we have introduced various paragraphs in our manuscript in accordance with the reviewer's comment.
2.
The second suggestion is related to the possibility of a geographical approach, as well as dealing with the different varieties of Olea europaea together with the dependent generation of secondary products. Likewise, with the comments in the previous paragraph, and this is not a justification but a reality, we are finishing the development of what we believe to be an interesting article related to these points mentioned and which we have done in collaboration with an excellent research group from the University of Jaén, the town where the highest percentage of olive oil is produced and where there are more varieties of olives in Europe. However, in this specific case, we believe that this aspect is somewhat distant from the objective of this review, which is simply to deal with the nutraceutical capacity of phenolic and triterpenic compounds during healthy growth.
We thank the Reviewer for all comments and insightful suggestions.
Reviewer 4 Report
Opinion
on the manuscript by Eva E. Rufino-Palomares et al., entitled "Role of Phytochemicals (Polyphenols and Triterpenes) present in Fruit and Olive Leaf Extracts as Antioxidants, Anti-infectives and Anticarcinogens in the Healthy Animal Growth"
Manuscript ID: molecules-1593262
The manuscript of Eva E. Rufino-Palomares et al. is a review giving a summary of the positive health effects of phytochemicals present in olive. The topic is outstandingly essential, and a well-structured and up-to-date review is undoubtedly helpful for the scientific community. But unfortunately the current review contains some elemental weaknesses which may hinder its publication in the present form. These are, among others, the following points:
Major points:
- The first point is the originality of the paper. Several excellent reviews have been published on the topic of the manuscript. Just a few examples: Jing Z et al: Review of the Biological Activity of Maslinic Acid. Curr Drug Targets 2021 doi: 10.2174/1389450122666210308111159; D'Angelo C et al: Wide Biological Role of Hydroxytyrosol: Possible Therapeutic and Preventive Properties in Cardiovascular Diseases. Cells 2020 doi: 10.3390/cells9091932; Rocha J et al: Table Olives and Health: A Review. J Nutr Sci 2020 doi: 10.1017/jns.2020.50, Martínez-Zamora L et al: Olive Tree Derivatives and Hydroxytyrosol: Their Potential Effects on Human Health and Its Use as Functional Ingredient in Meat. Foods 2021 doi: 10.3390/foods10112611. The manuscript's added value seems limited based on these and other reviews.
- The text contains some obvious statements. E.g., lines 59-60: "Proper nutrition is arguably the cornerstone of preventive health" line 166: "The bioactivities of HT are a consequence of its structure." This latter is valid for all biologically active molecules. Also, line 246: "Nowadays, cancer is among the deadliest diseases." These sound dissonant in a high-quality review.
- Lines 268-270: "Other authors also studied HL60 cells, showing that HT inhibits cell proliferation, blocking the G1 phase of the cycle, with a proportional increase of cells in the G0/G1 phase and a concomitant decrease in the S and G2/M phases [30]." The sentence is confusing since G1 and G0/G1 refer to the same cell population in the cell cycle analysis.
- Line 271: "HT (100 mM) causes an increase" but the authors of the cited reference used 100 microM instead of 100 mM.
- There is a hint to the anti-angiogenic capacity of HT (line 289) in the summarizing paragraph, but this property is not mentioned before, and there is no citation supporting that.
- Lines 325-326: "In vivo studies in rats with acute inflammation induced by intravenous injection of different doses of carrageenan showed that …". Carrageenan is not used intravenously. Instead, it is typically used as an intraplantar injection, as described in the cited reference.
- Lines 474-477: “In a recent study taking advantage of the anti-inflammatory capacity of triterpenes, Nagai et al. in an excellent clinical trial (CT) [77] have shown that MA intake is able to reduce proinflammatory cytokines, which are involved in muscle fibre atrophy, improving muscle response to resistance training through anti-inflammatory action.” There is no cytokine determination in the cited paper [77]. Instead, the authors of [77] cited some previous results about citokines.
Minor points:
- What is the exact meaning of "novel biological effects"? (line 92)
- The abbreviation HT for hydroxytyrosol is defined in line 118, but the full name is used again in line 134. A second definition, "Hydroxytyrosol (HT)" is given again in line 160.
- The precise meaning of dose-dependent absorption (lines 184-186) is unclear. May it refer to nonlinear kinetics?
- The way of writing the titles of cited papers is not consistent. In some references, all the title words are started with capital, in others only the first word. A careful check is suggested.
Altogether, the manuscript contains several mistakes or misleading points. Based on all these objections, the rejection of the current version of the review is suggested. Therefore, reconsideration seems rationale after a substantial improvement.
Author Response
REVIEWER 4
Article: molecules-1593262
Title: Role of Phytochemicals (Polyphenols and Triterpenes) present in Fruit and Olive Leaf Extracts as Antioxidants, Anti-infectives and Anticarcinogens in the Healthy Animal Growth
Authors: Rufino-Palomares, E.E. et al.
General comments:
First of all, the authors of the present review would like to deeply thank the different reviewers of the paper, for the time dedicated to the review of the paper for two fundamental reasons; the first reason, because of the absolute necessity that all scientific papers submitted for publication must be endorsed and accepted after peer review, as that is the only way to ensure their scientific potential and thus be able to advance, in an adequate and safe way, science. The second reason is that the criticisms made by peer review can significantly improve the quality of the work itself.
Secondly, we would also like to thank the different reviewers for both the good criticisms made of our work and the less favourable ones, as we understand that both types of criticisms undoubtedly help to increase the quality of the work. We believe that all the comments are worth taking into account and we are going to reply to them point by point and to modify and/or qualify everything that we believe may serve to improve the review and are in accordance with the objectives set out in the review.
RESPONSES TO REVIEWER 4
Major points
1.- It is true, in agreement with the reviewer, that when a scientific topic is interesting from an academic and health point of view, reviews on this topic abound. However, all published and unpublished works are of high interest as each one contributes specific and interesting aspects due to the special bias that each one of them brings to the scientific community. In our case, we respectfully disagree with the reviewer as our review brings specific and differential aspects to the rest of the reviews and especially to those commented by the reviewer. One fact that demonstrates what we say is that in our review the latest findings related to the bioactivities of these phytochemicals have been updated in a real way and as an example of this we give four brief reasons:
a) More than 25% of the bibliographic citations in this study correspond to the last five years and that almost 5% correspond to the years 2021-2022.
b) Our review covers not only HT as a polyphenol and MA as a triterpene, but also extends to a larger number of these classes of phytochemicals.
c) The main focus of our study is on the nutraceutical role of these compounds in healthy animal growth.
d) Our review includes such novel aspects, from a molecular and regulatory point of view, that the effects of some of these compounds on the mediation of their bioactivities by microRNAs are addressed.
2.- Thank you for your comment. The reviewer is quite right about the obviousness of certain phrases that are sometimes used to emphasize a certain point, so we have removed them from the text.
3.- Thank you for your insightful observation. The reviewer is absolutely right. In accordance with your comment, we have corrected the error we made when analysing the Fabiani et al. paper and have rewritten the paragraph and as such it is in red in the revised version.
4.- The reviewer is absolutely right. According to his comment we have corrected the error and as such it is in red in the revised version.
5.- Again the reviewer is absolutely right and we appreciate his comment. Reviewing our notes, we have observed that a paragraph dedicated to this ability was not added to the text. In the revised version we have added that paragraph and the supporting quote in red. Thanks again for your observation.
6.- Thanks again for your interesting observation. The authors of the work do not adequately specify the form of treatment of this compound in the material and methods section and since our group was not aware of this methodology, we assumed that it was that way. A rereading of the article in question indicates that the inflammation corresponds to the hind legs and therefore allows us to understand the reviewer's point and that this treatment is done through interplantar injection. In the revised version and in red we have corrected and I hope that this time the information is correct.
7.- We again appreciate the pertinent observation made by the reviewer on this point. Indeed, the anti-inflammatory response of maslinic acid in improving muscle response in older people to prevent mobility-related disability in these people was not the subject of this study, although the authors do directly relate this capacity to the positive effects on the parameters measured in the clinical trial. Therefore, we have corrected the paragraph in this sense and the modifications appear in red in the revised version.
Minor points
According to the reviewer, the first three issues raised have been corrected in the revised version of the original manuscript.
Regarding the fourth, we have simply transferred the title of the different papers as they appear in the different journals. We also wanted to check whether this way of expressing citations was included in the journal's rules and we have observed that the way we have put them in the text appears in most of the papers published in the journals of the MDPI group. However, if the reviewer and the technical team of the journal require it, we are willing to modify the entire bibliography in accordance with the rules that they tell us to do so.
We thank the Reviewer for all comments and insightful suggestions.
Round 2
Reviewer 2 Report
Reviewer Report of molecules-1593262-R2
Comments to Authors
The review articles describe "Role of Phytochemicals (Polyphenols and Triterpenes) present in Fruit and Olive Leaf Extracts as Antioxidants, Anti-infectives, and Anticarcinogens in the Healthy Animal Growth". As shown in the manuscript, the main objective of this review article wants to shed light on the role of polyphenols and triterpenes in healthy animal growth. Overall, some comments suggest the authors should be explained them well.
Comments from reviewer
- Comment 3 on Reviewer Report of molecules-1593262
The review article included OLIVEMAS® and OLIVESAN® but did explain any essential pieces of information. However, they are trademark products in the markets. It seems to have some business behavior.
Response 3
The fact that this review deals with the nutraceutical role of the components of olive tree extracts on healthy growth has motivated us to include two representative tables of two types of enriched extracts, one in polyphenols and the other in triterpenes, which could serve as a nutritional supplement. In our geographical area there is a company dedicated to the production of different types of natural extracts used in nutrition. The company is Extractos y Derivados, S.L. (Granada, Spain) as described at the bottom of both tables.
In agreement with the reviewer and in order not to interfere with any kind of product branding or commercial competition we can remove the trademarks from the title of both tables.
Comment form reviewer
The information in Tables 1 and 2, is still not related to the goal of the title of this manuscript.
- Comment 4 on Reviewer Report of molecules-1593262
Figures 3 and 5 still have inaccurate descriptions, and many signaling pathways were wrong and lacked enough pieces of information to support them.
Response 4
The reviewer is most certainly referring to figures 3 and 5 where we illustrate the main intracellular signalling mechanisms triggered by the different components of olive extracts, polyphenols and triterpenes. It is not true that the mechanisms described in the figures are not properly contrasted in the information implicit in the manuscript. Precisely, not all of them are described in the text, but the most significant ones are.
These figures do not attempt to report the individualised mechanisms of each of the polyphenolic or triterpene components analysed in this review; they are, in fact, a global and schematic summary of most of the signalling pathways set in motion by these components as a whole, and what is certain is that all of them are duly described throughout the text of the manuscript.
Comment form reviewer
- In Figure3, the authors still cannot support any pieces of evidence to support those polyphenols that could transport into the cell membrane.
- Moreover, In Figure 5, the authors also cannot support any pieces of evidence to support those triterpenoids that could have activated the receptor on the cell membrane.
- In Figures 3 and 5, the signal pathways still have inaccurate descriptions, and many signaling pathways were wrong and lacked enough pieces of information to support them. Suggest the authors should be explained them well.

Reviewer 4 Report
Some of the previously raized points have not been corrected. E.g., "HT (100 mM) causes an increase" but the authors of the cited reference used 100 microM instead of 100 mM. This sentence is a misleading statement. Chapter "2.1. Pharmacokinetics and toxicity": "following hyperbolic kinetics." The exact meaning of the term "hyperbolic kinetics" is not clear.
In my opinion, the improvement is not substantial enough and, therefore, I cannot consider the revised version acceptable.